# FreqExit: Enabling Early-Exit Inference for Visual Autoregressive Models via Frequency-Aware Guidance

**Ying Li**[1]    **Chengfei Lv**[2]    **Huan Wang**[1*]

[1]Westlake University    [2]Alibaba Group

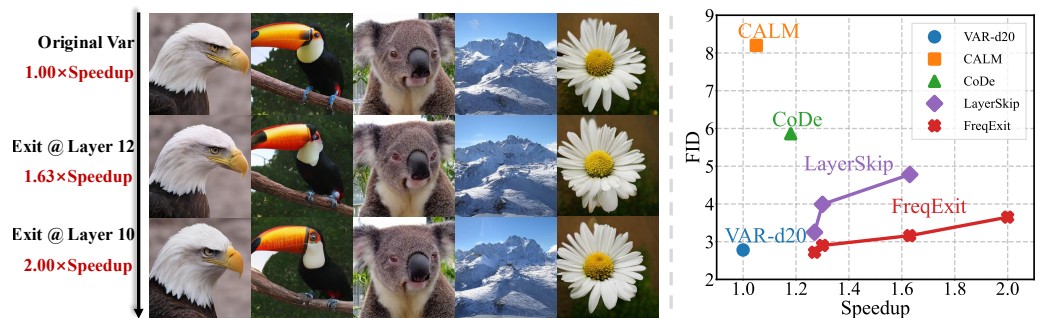

Figure 1: We introduce **FreqExit**, a dynamic inference strategy for next-scale visual autoregressive generation with a proposed frequency-aware guidance, bridging the gap between step-wise generation and early-exit-based acceleration. It achieves up to **2× speedup** with negligible quality degradation, offering a superior trade-off between efficiency and fidelity compared to prior baselines.

## Abstract

Visual AutoRegressive (VAR) modeling employs a next-scale decoding paradigm that progresses from coarse structures to fine details. While enhancing fidelity and scalability, this approach challenges two fundamental assumptions of conventional dynamic inference: semantic stability (intermediate outputs approximating final results) and monotonic locality (smooth representation evolution across layers), which renders existing dynamic inference methods ineffective for VAR models. To address this challenge, we propose *FreqExit*, an integrated loss design that enables dynamic inference in VAR without altering its architecture or compromising output quality. FreqExit is based on a key insight: high-frequency details are crucial for perceptual quality and tend to emerge only in later decoding stages. Leveraging this insight, we design targeted mechanisms that guide the model to learn more effectively through frequency-aware supervision. The proposed framework consists of three components: (1) a curriculum-based supervision strategy with progressive layer dropout and early exit loss; (2) a wavelet-domain high-frequency consistency loss that aligns spectral content across different generation steps; and (3) a lightweight self-supervised frequency-gated module that guides adaptive learning of both structural and detailed spectral components. On ImageNet 256×256, FreqExit achieves up to **2×** speedup with only minor degradation, and delivers **1.3×** acceleration without perceptible quality loss. This enables runtime-adaptive acceleration within a consistent design tailored for next-scale VAR, offering a favorable trade-off between efficiency and fidelity for practical and flexible deployment. Code is available at https://github.com/NeuraLiying/FreqExit.

---

*Corresponding author. Email: wanghuan@westlake.edu.cn

39th Conference on Neural Information Processing Systems (NeurIPS 2025).

# 1 Introduction

Autoregressive models [1, 2, 3, 4, 5, 6, 7] have achieved widespread success, but their substantial computational and memory demands during inference [8, 9, 10, 11, 12] pose major obstacles to deployment in resource-limited environments [13, 14]. To address this challenge, dynamic inference has emerged as a promising strategy for accelerating large models during deployment, enabling the model to adjust its computational path based on the input [15, 16, 17, 18]. This flexibility allows the model to retain full capacity for complex inputs while reducing computation for simpler ones, supporting a flexible trade-off between performance and efficiency. With the development of visual autoregressive models such as LLama-Gen [19] and VAR [20], their capacity to handle complex vision tasks has attracted increasing attention, motivating efforts to accelerate inference in this domain [21, 22, 23, 24]. VAR has gained popularity for its next-scale prediction paradigm, which replaces token-by-token generation with hierarchical token map decoding. This generation paradigm has gained increasing popularity for its ability to improve inference efficiency and scalability, while maintaining high-quality image synthesis. However, the application of dynamic inference to VAR presents unique challenges. The decoding paradigm based on token maps in VAR breaks the granularity assumptions behind token-level control methods such as speculative decoding [17, 25]. In addition, our analysis (Sec. 3.1) shows that transformer layer representations remain highly dynamic and non-redundant, which invalidates the assumptions of layer stability required by early-exit [26, 27] and layer compression [28, 29] approaches designed for token-wise models.

To address this challenge, we perform an in-depth analysis of the VAR generation dynamics and identify distinct frequencies domain variations throughout the decoding steps. This progressive shift from low-frequency structures in the early steps to high-frequency details in later steps leads to step-dependent feature semantics. These step-dependent frequency shifts and coarse-to-fine token map generation patterns lead to unstable intermediate representations and limited predictability, which undermine the effectiveness of conventional dynamic inference strategies. Based on these insights, we propose FreqExit, a cohesive training framework that integrates three synergistic mechanisms to support efficient and adaptive inference in VAR: (1) a curriculum-based supervision strategy that progressively activates intermediate layer training, (2) a frequency-aware consistency loss that guides high-frequency reconstruction at later steps, and (3) a frequency-gated self-reconstruction loss that adaptively regularizes spectral learning during training. Together, these components robustly enhance intermediate representations and enable efficient early exits without compromising generation quality. Experimental results demonstrate the efficiency of our method. As shown in Figure 1, our method achieves up to 2× acceleration with minor quality degradation, and delivers 1.3× speedup with no perceptible loss—offering flexible trade-offs between efficiency and overall generation fidelity. Our main contributions are summarized as follows:

- We propose an integrated training framework for accelerating dynamic inference in visual autoregressive models, specifically tailored to the next-scale generation paradigm of VAR.

- We introduce two training-time components: a step-wise high-frequency consistency loss and a lightweight frequency-gated self-reconstruction (FGSR) module, which improve representation quality and accelerate training without incurring any inference-time overhead.

- Extensive experiments demonstrate that our method achieves up to 2× inference speedup with negligible quality degradation. To the best of our knowledge, this is the first work to enable early-exit-based dynamic inference in next-scale generation models like VAR.

# 2 Related Work

## 2.1 Dynamic Inference Methods

Dynamic inference refers to the ability of a model to adjust its computational pathway based on input, allowing selective execution of layers, channels, or tokens to improve inference efficiency [30, 31, 32]. Compared to static compression methods, dynamic approaches offer better adaptability across inputs of varying complexity. Early works have explored the depth of dynamic layers, channel width, and routing mechanisms [16, 33, 34, 35]. More recent research extends this idea to transformer-based architectures in both language and vision domains, where dynamic inference is used to adaptively control the depth of decoding or the granularity of token-level computation [27, 28, 36, 37, 38].

**Early Exiting** As a representative form of dynamic inference, early exiting enables a model to terminate inference at intermediate layers based on signals such as confidence or consistency [26, 36, 39, 40]. Sample-level approaches determine the required depth based on overall input complexity, typically by attaching lightweight classifiers [41, 42] or monitoring output consistency across layers [39, 43]. At the token level, early exiting has been applied to both encoder and decoder architectures. CALM [26] uses confidence-based classifiers and handles KV cache inconsistencies via hidden state reuse. BERxiT [44], BE3R [45], and EE-LLM [46] extend token-level exits using learned classifiers. SkipDecode [36] improves batch-level efficiency by enforcing unified and monotonic exits. Raposo et al. [28] introduce trainable routers for dynamic depth control. Layer-Skip [27] eliminates explicit confidence estimation by unifying early-exit training with speculative decoding. Recent efforts also extend early exit to token-wise visual generation, including MuE [40], AdaNAT [47], and DeeR-VLA [38].

## 2.2 Inference Acceleration for Visual Generation

Autoregressive image generation has evolved from early raster-scan models [48, 49] to transformer-based frameworks that sequentially generate discrete image tokens via VQVAE [50] or VQGAN [51]. Recent models such as LlamaGen [19] and MAR [52] achieve strong results in high-resolution and multimodal generation [53, 54]. VAR [20] introduces a next-scale paradigm that outputs entire token maps per step. This hierarchical design enhances scalability and has since inspired a wide range of follow-up works spanning different modalities and tasks [55, 56, 57, 58, 59, 60].

Despite the rapid progress in visual autoregressive modeling, research on inference acceleration remains at an early stage. Several approaches have been proposed to reduce decoding latency and computational cost. Token-parallel acceleration methods such as speculative decoding [23, 25] aim to predict multiple tokens simultaneously or restructure the inference schedule for faster generation. However, these methods are fundamentally incompatible with next-scale generation models like VAR, as the latter produce entire token maps per step instead of single-token outputs, undermining the assumptions of token-level parallelism. CoDe [61] proposes a collaborative decoding strategy to accelerate VAR by assigning large and small models to different scales. Nonetheless, it relies on separately fine-tuned models with fixed inference paths, and its dependence on a large backbone limits applicability in resource-constrained settings where a single lightweight model is preferred. To address these limitations, we introduce a dynamic inference framework tailored for the hierarchical next-scale generation paradigm, enabling efficient acceleration through early exits.

# 3 Method

## 3.1 Comprehensive Analysis of VAR Generation Dynamics

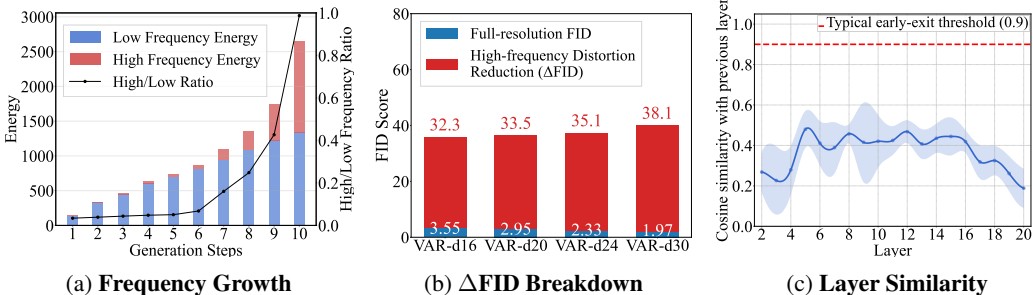

(a) **Frequency Growth**      (b) **ΔFID Breakdown**      (c) **Layer Similarity**

Figure 2: Comprehensive visualization of VAR generation dynamics. (a) Frequency component evolution across steps in VAR-d20. (b) FID degradation caused by high-frequency removal. (c) Cosine similarity between token maps from consecutive layers.

To enable dynamic acceleration tailored for next-scale generation, we analyze the generative behavior of VAR [20] through discrete wavelet transform (DWT)-based frequency decomposition, a technique shown effective in prior generative model studies [62, 63], to reveal its step-wise spectral characteristics. Specifically, we decompose token maps at each generation step into low- and high-frequency components, and study their evolution across steps and layers. We further compare model variants (d16, d20, d24, d30) to assess how high-frequency removal affects generation quality. In parallel, we

conduct layer analysis by recording the output token map at each transformer block and computing its cosine similarity with the previous layer, quantifying how representations shift throughout the network. Fig. 2 illustrates three key findings that characterize the generative behavior of VAR.

***Observation 1.*** **Frequency progression across steps**   As shown in Fig. 2a, the first six steps predominantly reconstruct low-frequency structures, while steps 7–10 introduce a sharp increase in high-frequency content. This step-wise divergence reveals a two-stage generation process, where the first steps establish the global structure and later steps refine fine-grained details, making it difficult for the model to maintain consistent representations across stages.

***Observation 2.*** **High-frequency accuracy drives quality**   All models experience substantial performance degradation when high-frequency details are removed, with the impact increasing with model size—highlighting that deeper models primarily improve visual quality by modeling fine details more precisely. This highlights the importance of high-frequency modeling for improving early-layer exit capability, as generative quality relies heavily on accurate reconstruction of fine textures rather than low-frequency structure alone.

***Observation 3.*** **No redundant layers**   We assess layer-wise changes using truncated inference and cosine similarity between successive token maps (Fig. 2c). Similarity values remain consistently low (ranging between 0.2–0.6), even at the final layers (e.g., between layer 19 and 20), suggesting that each transformer block continues to modify the representation meaningfully. This contradicts the layer stabilization assumption commonly adopted in early-exit strategies such as CALM [26], leaving no obvious "safe" point for early termination.

## 3.2   Our Method

**Motivation.** Our analysis (Sec. 3.1) reveals that VAR follows a hierarchical next-scale decoding paradigm, where generation gradually shifts from low-frequency structures to high-frequency details. This results in step-wise frequency transitions and unstable inter-layer representations that hinder reliable early-exit decisions. In contrast to conventional autoregressive models, intermediate outputs in VAR are highly dynamic and lack semantic stability, as tokens are repeatedly refined across steps rather than converging to fixed representations. Accurately modeling high-frequency components therefore becomes essential for maintaining output quality during early termination.

To address these challenges, we propose an integrated training framework that restructures intermediate representations to support early exit without degrading output quality. It integrates: (1) curriculum-based supervision to enhance shallow layers, (2) a frequency-aware consistency loss to align multi-step predictions, and (3)a frequency-gated self-reconstruction (FGSR) module for modeling cross-band dependencies. An overview of the proposed framework is illustrated in Fig. 3.

### 3.2.1   Curriculum-Based Early-Exit Supervision

We adopt a curriculum learning-based early-exit training framework that jointly supervises intermediate layers to support efficient dynamic inference, shown in Fig. 3(a). It combines depth-aware layer dropout with cross-entropy loss at each supervised layer, and further introduces a lightweight, layer-adaptive distillation mechanism that adjusts supervision strength by depth.

**Layer Dropout.**   To encourage shallow layers to learn stronger representations, we apply layer dropout during training by randomly skipping layers with depth-dependent probabilities. Specifically, deeper layers are dropped more frequently to promote early-layer expressiveness. The hidden state update for layer $\ell$ at iteration $t$ is:

$$x_{\ell+1,t} = x_{\ell,t} + M\big(p_{\ell,t}\big) f_\ell(x_{\ell,t}), \tag{1}$$

where $M(p)$ is a Bernoulli mask, and the dropout probability $p_{\ell,t}$ increases with depth.

**Early-Exit Loss.**   We apply an early-exit loss to intermediate layers, enabling dynamic inference without modifying the architecture. Specifically, each transformer block is connected to a shared LM

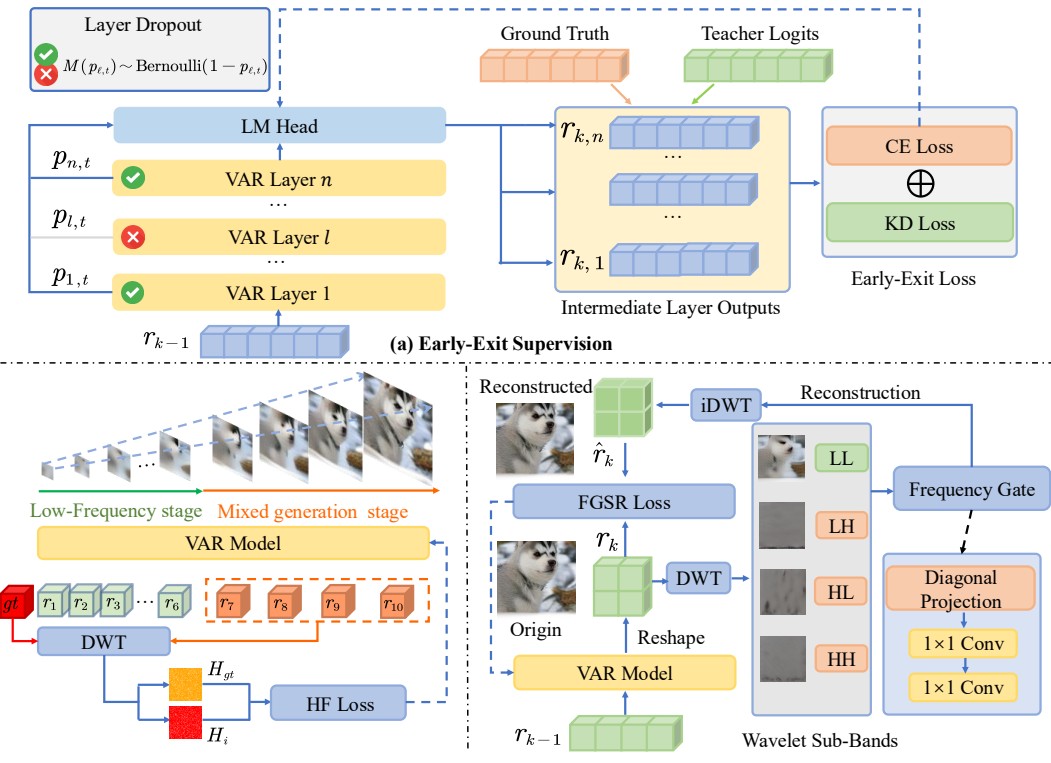

**(a) Early-Exit Supervision**

**(b) Progressive High-Frequency Consistency Loss**

**(c) Auxiliary Frequency-Gated Self-Reconstruction Loss**

Figure 3: **Overview of our proposed FreqExit framework. (a) Curriculum-Based Early-Exit Supervision:** integrates depth-aware layer dropout and adaptive early-exit loss with layer-specific knowledge distillation, encouraging shallow layers to learn expressive features under a dynamic supervision curriculum. **(b) High-Frequency Consistency Loss:** enforces step-wise spectral alignment between student and teacher predictions in the wavelet domain, stabilizing high-frequency learning across generation steps without disrupting early training behavior. **(c) Frequency-Gated Self-Reconstruction Loss:** provides an auxiliary loss branch with learnable sub-band gates that modulate wavelet components and guide the model toward frequency-aware spectral reconstruction, improving convergence and generation quality without affecting inference runtime.

head $g$ and supervised by a cross-entropy loss:

$$\mathcal{L}_{\text{EE}} = \sum_{\ell=0}^{L-1} \tilde{e}(t,\ell)\, \mathcal{J}_{\text{CE}}\big(g(x_{\ell+1,t}), Y\big), \quad \tilde{e}(t,\ell) = \frac{C(t,\ell)\, e(\ell)}{\sum_{i=0}^{L-1} C(t,i)\, e(i)}. \quad (2)$$

Here $C(t,\ell)$ is a binary curriculum gate that activates exits gradually, while $e(\ell)$ increases quadratically with depth to place more weight on later layers.

To improve learning at shallow exits, we introduce a layer-adaptive knowledge distillation loss that provides softened teacher guidance with temperature annealing:

$$\mathcal{L}_{\text{KD}} = \frac{1}{\sum_{\ell \in \mathcal{S}} w_\ell} \sum_{\ell \in \mathcal{S}} w_\ell\, T_\ell^2\, \text{KL}\left(q_t/T_\ell \,\big\|\, q_s^\ell/T_\ell\right), \quad T_\ell = T_{\max} - (T_{\max} - T_{\min})\frac{\ell}{L}, \quad (3)$$

where $q_t$ and $q_s^\ell s$ denote the teacher (full VAR-d20) and layer-wise student logits, respectively, and the temperature $T_\ell$ decreases from $4.0$ (shallow layers) to $1.0$ (deep layers), providing softer targets to stabilize shallow-layer training, where semantic information is limited. The early-exit loss combines $\mathcal{L}_{\text{EE}}$ and $\mathcal{L}_{\text{KD}}$, guiding intermediate exits under the curriculum schedule.

**Curriculum Scheduling.** To ensure stable training and effective supervision, we adopt a rotational curriculum $C(t,\ell)$ that activates early-exit loss on a subset of layers at each iteration. Specifically, a fixed number of layers are selected in a round-robin manner, allowing all layers to be periodically updated while avoiding conflicting gradient signals between shallow and deep layers. This strategy

mitigates the dominance of gradients in deeper layers and promotes more balanced learning across depths. Details of the curriculum schedule for $C(t, \ell)$ and $S(t)$ are provided in Appendix A in the supplementary material.

### 3.2.2 High-Frequency Consistency Loss

As shown in Fig. 3(b), we introduce a high-frequency consistency loss to guide intermediate representations during training. While high-frequency components are essential in the later stages of next-scale decoding (Sec.3.1), they are often underrepresented in earlier layers due to resolution mismatch and immature semantics. Direct supervision at these stages may destabilize training. To mitigate this, we propose a progressive loss that softly aligns student and teacher predictions in the wavelet domain through step-wise and epoch-wise scheduling.

For each layer $\ell$ and generation step $n$ with $p_n \times p_n$ patches, we reshape the token map into a spatial grid and apply a 2D Haar DWT:

$$\{H_n(\ell),\ L_n(\ell)\} = \text{DWT}\big(\text{reshape}(x_{\ell,n})\big), \tag{4}$$

and extract the high-frequency bands $H_n^s(\ell)$ and $H_n^t$ from student and teacher. To balance gradient contributions across resolutions, we apply step-aware weights $w_n$:

$$w_n = \left(\frac{p_n}{p_{\max}}\right)^\gamma, \tag{5}$$

where $\gamma$ is a tunable scaling factor. The total high-frequency consistency loss is defined as:

$$\mathcal{L}_{\text{HF}} = \frac{1}{\sum_{n \in \mathcal{P}} w_n} \sum_{\ell \in \mathcal{S}} \sum_{n \in \mathcal{P}} w_n \left\| H_n^s(\ell) - H_n^t \right\|_2^2. \tag{6}$$

To prevent premature constraints on insufficiently trained layers, $\mathcal{L}_{\text{HF}}$ is initially disabled and gradually activated as training progresses. A dynamic scaling mechanism adjusts its contribution based on the moving average ratio between $\mathcal{L}_{\text{HF}}$ and other losses, ensuring stability throughout.

### 3.2.3 Frequency-Gated Self-Reconstruction Loss

A lightweight **Frequency-Gated Self-Reconstruction (FGSR)** module is proposed to introduce a residual training path with learnable gates over wavelet sub-bands, as shown in Fig. 3(c). Unlike traditional auxiliary branches, FGSR provides interpretable frequency-aware supervision while leaving the main architecture unchanged. Frequency-aware optimization during next-scale generation is achieved by incorporating a learnable gating mechanism over wavelet sub-bands. Let $r_t$ denote the token map at generation step $t$, and $r_{\text{aux},t}$ the auxiliary output reconstructed from gated frequency components. The training objective is:

$$\mathcal{L}_{\text{total}} = \mathcal{L}_{\text{CE}}(r_t) + \lambda \cdot \left\| r_{\text{aux},t} - r_t \right\|_2^2, \tag{7}$$

where $\mathcal{L}_{\text{CE}}$ is the main task loss and $\lambda$ controls the strength of reconstruction guidance. Each sub-band $b \in \{LL, LH, HL, HH\}$ is modulated by a learnable gate $\gamma_b$ as $b' = b \cdot e^{\gamma_b}$. The gradient of $\mathcal{L}_{\text{total}}$ w.r.t. $\gamma_b$ comprises components from both the main loss and reconstruction loss. Under the assumption $r_t = \text{iDWT}(\cdots, b \cdot e^{\gamma_b}, \cdots)$, the regularizing term becomes:

$$\frac{\partial \| r_{\text{aux},t} - r_t \|_2^2}{\partial \gamma_b} = 2(\gamma_b - 1) \cdot \|b\|_2^2, \tag{8}$$

which pulls $\gamma_b$ toward 1 unless overridden by task gradients. At convergence, the closed-form update that balances task alignment and reconstruction stability is:

$$\gamma_b = 1 + \frac{1}{\lambda} \cdot \frac{\left\langle \frac{\partial \mathcal{L}_{\text{CE}}}{\partial r_t}, b \right\rangle}{\|b\|_2^2}. \tag{9}$$

Gates $\gamma_b$ are enhanced when their frequency band $b$ aligns with task gradients, guiding the model from low-frequency structure to high-frequency detail over time. This behavior is realized through the following frequency-gated reconstruction process.

FGSR operates on intermediate feature maps $r \in \mathbb{R}^{B \times C \times H \times W}$, where $C$ denotes the number of channels. Each feature map is reshaped into a 2D spatial layout and decomposed via discrete wavelet transform (DWT) into four frequency sub-bands. A learnable exponential gate $\gamma_b$ is applied to each sub-band, yielding the modulated band $b' = b \cdot e^{\gamma_b}$. The gated sub-bands are concatenated and projected through a shared linear transformation $W_{\text{fgp}} \in \mathbb{R}^{C \times 4C}$, then reconstructed via its transpose and inverse DWT:

$$\hat{r} = \text{iDWT}\left(W_{\text{fgp}}^{\top} W_{\text{fgp}} [LL', LH', HL', HH']\right). \tag{10}$$

The frequency-gated reconstruction loss is defined over generation steps $\mathcal{A}$ and supervised layers $\mathcal{S}$:

$$\mathcal{L}_{\text{FGSR}} = \frac{1}{|\mathcal{S}|} \sum_{\ell \in \mathcal{S}} \frac{1}{|\mathcal{A}|} \sum_{i \in \mathcal{A}} \left\| f_{\text{FGSR}}(r_i^{\ell}) - r_i^{\ell} \right\|_2^2 + \lambda \cdot \mathcal{L}_{\text{align}}, \tag{11}$$

where $f_{\text{FGSR}}$ denotes the reconstruction function, and $\lambda$ balances the auxiliary loss. A projection alignment regularizer is introduced to promote numerical stability and reversibility, where $\text{Sym}(A) = \frac{1}{2}(A + A^{\top})$ is the symmetrization operator, and $W_{\text{inv}}$ approximates $W_{\text{fgp}}^{\top}$:

$$\mathcal{L}_{\text{align}} = \left\| \text{Sym}(W_{\text{fgp}}^{\top} W_{\text{fgp}} - I) \right\|_F^2 + \left\| W_{\text{inv}} - W_{\text{fgp}}^{\top} \right\|_F^2, \tag{12}$$

FGSR serves as a *lightweight*, *inference-free* auxiliary branch designed to address the mismatch in generation patterns across different stages. Early steps tend to produce low-frequency structures, while later steps emphasize both low- and high-frequency details. This shift poses challenges for joint training across different stages. FGSR resolves this by introducing frequency-gated supervision that softly guides optimization without interfering with the main gradient flow. Specifically, when the reconstruction loss remains small, the model maintains emphasis on low-frequency components. As high-frequency details become important, the loss increases, prompting the gates to adaptively adjust their influence—thereby signaling the model to refine its spectral representation. Through this loss-driven feedback loop, FGSR enables *stage-aware*, *spectrum-adaptive* training that improves convergence without modifying architecture. The specific implementation details and pseudocode of FGSR can be found in Appendix A of the supplementary material.

**Overall Training Objective.** Following the integration of all supervision components, the final training objective of FreqExit is formulated as:

$$\mathcal{L}_{\text{total}} = \lambda_1 \mathcal{L}_{\text{EE}} + \lambda_2 \mathcal{L}_{\text{KD}} + \lambda_3 \mathcal{L}_{\text{HF}} + \lambda_4 \mathcal{L}_{\text{FGSR}}, \tag{13}$$

where $\mathcal{L}_{\text{EE}}$ and $\mathcal{L}_{\text{KD}}$ jointly constitute the early-exit supervision loss, providing primary learning signals for intermediate layers, while $\mathcal{L}_{\text{HF}}$ and $\mathcal{L}_{\text{FGSR}}$ form the frequency adaptation loss that enhances spectral consistency and improves high-frequency reconstruction quality. The weighting coefficients $\lambda_i$ control the relative contribution of each component and are provided in Appendix C.

## 4 Experimental Results

### 4.1 Experimental Setup

**Baseline Methods.** We evaluate our method in comparison with several representative baselines. **CALM** [26] is a training-free method that exits based on the softmax gap between top-1 and top-2 logits. As the default threshold of 0.9 rarely triggers exits under layer-wise dynamics of VAR, it is lowered to 0.6 to enable meaningful comparison. **LayerSkip** [27] enables flexible truncation via shared early-exit loss and layer dropout. Its layer-wise mechanism is adapted to VAR, without speculative decoding, which is incompatible with scale-wise generation. **CoDe** [61] adopts a two-model collaboration between a drafter and a refiner. In our setup, VAR-d20 and VAR-d16 are used as the drafter and refiner, enabling step-wise refinement during inference. However, CoDe relies on two independently trained models with fixed inference paths, whereas FreqExit enables adaptive acceleration within a single model without architecture modification.

**Implementation Details.** All models are trained on ImageNet-1K [64]. CALM requires no training. LayerSkip and CoDe are trained for 80 epochs using AdamW with a batch size of 1024. Our method uses the same training setup as LayerSkip, with a learning rate of 1e-5 and no weight decay for 80 epochs. For CoDe, we evaluate both training-free and fine-tuned variants. In the fine-tuned setting, the drafter (VAR-d20) is trained for 15 epochs (learning rate 1e-6, weight decay 0.08), and the refiner

Table 1: Comparison of inference efficiency and generation quality across methods. **Param**: active parameters; **Layers**: average number of transformer layers used; **Speedup**: latency-based acceleration relative to VAR-d20; **Latency** (s): per-image inference time; **Thrpt** (it/s): throughput, i.e., images processed per second; **GFLOPs**: total floating-point operations; **FID**, **IS**: standard metrics for generation quality; **Prec.**: Precision; **Rec.**: Recall.

| Method | Param | Layers | Speedup ↑ | Latency ↓ | Thrpt ↑ | GFLOPs ↓ | FID ↓ | IS ↑ | Prec. ↑ | Rec. ↑ |
|---|---|---|---|---|---|---|---|---|---|---|
| VAR-d20 | 600M | 20 | 1.00× | 0.265 | 3.80 | 1879 | 2.78 | 252 | 0.84 | 0.55 |
| CALM | 546M | 18.2 | 1.05× | 0.252 | 3.07 | 1779 | 8.20 | 249 | 0.68 | 0.60 |
| CoDe (w/o training) | 900M | 18 | 1.18× | 0.224 | 4.66 | 1440 | 9.35 | 251 | 0.79 | 0.52 |
| CoDe (w/ training) | 900M | 18 | – | – | – | – | 5.86 | 252 | 0.80 | 0.54 |
| LayerSkip[a] | 480M | 16 | 1.27× | 0.209 | 4.36 | 1664 | 3.25 | 253 | 0.80 | 0.58 |
| LayerSkip[b] | 420M | 14 | 1.30× | 0.204 | 4.92 | 1466 | 3.99 | 251 | 0.78 | 0.57 |
| LayerSkip[c] | 360M | 12 | 1.63× | 0.162 | 6.21 | 1448 | 4.78 | 250 | 0.76 | 0.60 |
| **FreqExit**[a] (Ours) | 480M | 16 | 1.27× | 0.209 | 4.36 | 1664 | 2.72 | 252 | 0.81 | 0.55 |
| **FreqExit**[b] (Ours) | 420M | 14 | 1.30× | 0.204 | 4.92 | 1466 | 2.90 | 250 | 0.78 | 0.57 |
| **FreqExit**[c] (Ours) | 360M | 12 | 1.63× | 0.162 | 6.21 | 1448 | 3.16 | 251 | 0.71 | 0.57 |
| **FreqExit**[d] (Ours) | **300M** | **10** | **2.00×** | **0.130** | **7.72** | **1341** | 3.58 | 251 | 0.73 | 0.56 |

*Note.* VAR generates images in 10 autoregressive steps. We evaluate four early exit configurations reflecting different cost-accuracy trade-offs, where each vector lists transformer layers used at each generation step.

(VAR-d16) is distilled for 65 epochs (learning rate 1e-5, no weight decay), following the original CoDe setup. In practice, FreqExit does not require retraining VAR from scratch. We initialize from the official VAR-d20 checkpoint (trained for 250 epochs on ImageNet-1K) and fine-tune for an additional 80 epochs (∼30% cost) to enable early-exit capability. Detailed training configurations are detailed in Appendix C of the supplementary material.

**Evaluation Metrics.** For each baseline method, we generate 50,000 images (50 per class) across 1,000 ImageNet-1K classes, using the same inference settings on a single RTX 4090 GPU. Efficiency is evaluated by latency, FLOPs (measured using the `Torch.profiler`), throughput, memory usage, active parameters, and transformer layers (Param and layers). Speedup is computed relative to VAR-d20. Generation quality is assessed using four standard metrics: FID, IS, Precision, and Recall.

## 4.2 Main Results

On the ImageNet 256×256 benchmark, our FreqExit framework achieves up to 2.0× acceleration with only minor quality degradation (FID increases from 2.78 to 3.58), and provides 1.3× speedup with no perceptible loss in generation quality (FID = 2.72). These results are achieved under different early-exit configurations, demonstrating the flexibility of the method to support run-time-adaptive inference with a single model, allowing dynamic selection of exit strategies without retraining or architectural changes. Compared to baseline methods including CALM, LayerSkip, and CoDe (see Table 1), FreqExit consistently delivers better efficiency–quality trade-offs. It achieves lower latency, fewer FLOPs, and higher throughput under comparable or improved fidelity. For example, at the most aggressive setting, FreqExit reaches a throughput of 7.72 it/s, outperforming all baselines. When compared to LayerSkip, our method shows clear advantages, especially at shallower configurations. As the number of active layers decreases, FreqExit maintains better generation quality with lower FID scores—highlighting its ability to preserve fidelity even under tight computational budgets.

Overall, these results confirm that FreqExit enables efficient, flexible, and high-quality autoregressive generation. It effectively bridges the gap between next-scale decoding and practical dynamic inference, offering a scalable solution for deployment under varying resource budgets.

## 4.3 Dynamic Inference Capability

To evaluate the dynamic inference capability of our method, we construct a series of early-exit strategies with different layer configurations across the 10 autoregressive generation steps. Each strategy generates 50,000 images on a single NVIDIA RTX 4090 GPU, and the results are summarized in Table 2. Progressively reducing the average number of layers leads to substantial efficiency gains.

Table 2: Early exit strategies used in dynamic inference. Each vector indicates the number of transformer layers used at each of the 10 generation steps.

| Strategy | Exit Layers at Steps | Avg. Params | Avg. Layers | Max Batch Size | FID ↓ |
|----------|---------------------|-------------|-------------|----------------|-------|
| Full | [20, ..., 20] | 600M | 20.0 | 80 | 2.71 |
| ① | [20,20,20,20,20,16,16,16,16,16] | 540M | 18.0 | 96 | 2.72 |
| ② | [16, ..., 16] | 480M | 16.0 | 96 | 2.72 |
| ③ | [16,16,16,16,16,12,12,12,12,12] | 420M | 12.0 | 120 | 2.90 |
| ④ | [16, ..., 12] | 360M | 12.0 | 120 | 3.08 |
| ⑤ | [16,12,12,12,12,10,10,10,10,10] | 330M | 12.0 | 128 | 3.27 |
| ⑥ | [16,12,10, ..., 10] | 300M | 10.0 | 130 | 3.37 |

Table 3: Ablation study on training efficiency. FID values at intermediate layers (Layer 8 and Layer 12) across training epochs, comparing models with and without HF loss and FGSR. The proposed modules lead to faster convergence and significantly lower FID, especially at shallower layers.

| Epoch | FID ↓ with exit at Layer 8 | | FID ↓ with exit at Layer 12 | |
|-------|---------------|--------------|---------------|--------------|
| | w/o HF+FGSR | w/ HF+FGSR | w/o HF+FGSR | w/ HF+FGSR |
| 40 | 22.73 | 13.20 | 6.87 | 4.59 |
| 48 | 20.53 | 12.54 | 5.52 | 4.86 |
| 56 | 19.40 | 11.74 | 5.31 | 4.36 |
| 64 | 18.52 | 10.90 | 5.21 | 3.92 |
| 72 | 18.78 | 10.51 | 5.25 | 3.48 |
| 80 | 17.38 | 10.02 | 4.78 | 3.26 |

Compared to the full inference baseline, the most lightweight configuration uses only the first 10 layers per step, reducing active parameters by 50% and increasing the maximum batch size by **1.63×** (from 80 to 132). This comes with only a modest drop in quality, as FID increases by just 0.94.

We analyze these results from two perspectives: First, dynamic configurations significantly reduce computation while preserving image quality. Strategies averaging 16 or more layers achieve FID scores comparable to the full model, demonstrating that early exits can reduce cost without sacrificing fidelity. The reduced memory usage also allows for larger batch sizes and improved throughput. Second, this dynamic behavior aligns well with the autoregressive nature of VAR, where token counts increase across steps. Early steps, with fewer tokens and lower-resolution representations, primarily capture low-frequency structures and can benefit from deeper layers. In contrast, later steps are more expensive due to larger token maps and can use shallower exits to control cost. This per-step flexibility enables fine-grained, step-aware adjustment of inference depth, allowing the model to adaptively balance quality and efficiency throughout the generation process.

## 4.4 Ablation Study

To evaluate the effectiveness of our proposed training enhancements, we conduct an ablation study comparing models trained with and without the HF loss and FGSR modules. In this study, we track the FID scores obtained under early exit at Layer 8 and Layer 12 across training epochs. The results, presented in Table 3, show that models equipped with these modules consistently achieve lower FID and converge more rapidly than their counterparts. The improvement is particularly pronounced when exiting at shallower layers (e.g., Layer 8), where the FID decreases more sharply and stabilizes at an earlier stage of training. These findings suggest that our method not only improves final performance, but also accelerates representation learning, especially in the early layers. This can be attributed to two key factors: (1) the HF loss provides explicit supervision in the frequency domain, helping the model adapt to different frequency characteristics across generation steps; and (2) the FGSR module delivers frequency-aware reconstruction signals that guide optimization toward step-specific generation behaviors. Together, these components mitigate the mismatch between model capacity and evolving generation patterns, enabling more efficient learning in shallow layers.

# 5 Conclusion

We present FreqExit, an integrated training framework that enables dynamic inference in next-scale visual autoregressive models. By thoroughly analyzing the spectral and representational behavior of VAR, we identified key obstacles to efficient early exit, such as frequency progression across steps and unstable intermediate representations. FreqExit addresses these challenges through three targeted modules: a curriculum-based early-exit supervision strategy, a progressive wavelet-based high-frequency consistency loss, and a lightweight frequency-gated self-reconstruction module. Extensive experiments on ImageNet 256×256 demonstrate that FreqExit achieves up to **1.3× speedup with no perceptible quality degradation**, and up to **2× acceleration** with only minor or negligible losses. This flexible trade-off highlights the adaptive nature of FreqExit, which can dynamically adjust computational depth to meet diverse efficiency requirements. Overall, FreqExit provides a practical and generalizable solution for efficient early-exit acceleration in next-scale generation models.

## Acknowledgements

This paper is supported by Young Scientists Fund of the National Natural Science Foundation of China (No. 62506305), Zhejiang Leading Innovative and Entrepreneur Team Introduction Program (No. 2024R01007), Key Research and Development Program of Zhejiang Province (No. 2025C01026), Scientific Research Project of Westlake University (No. WU2025WF003), Chinese Association for Artificial Intelligence (CAAI) & Ant Group Research Fund - AGI Track (No. 2025CAAI-ANT-13), and the Special Support Talents Program of "Xi Hu Ming Zhu Program" in Hangzhou.

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

# FreqExit: Enabling Early-Exit Inference for Visual Autoregressive Models via Frequency-Aware Guidance

## Appendix

This appendix provides additional details on the proposed methodology, training configurations, and experimental results. Section A elaborates on implementation components such as early-exit supervision, frequency-gated reconstruction, and gradient-based loss tuning. Section C outlines training setups and hyperparameters for baseline comparisons. Section D presents qualitative visualizations under conditional generation and zero-shot inpainting. Section E concludes with a discussion of limitations and future directions, along with information for reproducibility.

## A Details of Methodology

### A.1 Curriculum-Based Early-Exit Supervision

**Depth-Aware Layer Dropout.** To promote the expressiveness of the shallow layer during training, we implement a depth-sensitive layer dropout strategy. Each transformer block $l$ is skipped independently with a depth-increasing probability:

$$p_l = \min\left(\max\left(\left(e^{\frac{\ln 2}{L-1} \cdot l} - 1\right), 0\right), 1\right) \cdot p_{\max},\tag{14}$$

where $L$ is the total number of layers and $p_{\max}$ is a global dropout scaling factor (default 0.1). This formulation ensures that early layers are less likely to be skipped during training, thereby encouraging them to learn stronger and more generalizable representations.

**Layer-Wise Exit Weights.** To prioritize deeper exits while still training shallow ones, we assign a normalized weight $w_l$ to the loss of each layer based on its relative depth. Let $\mathcal{S} \subseteq 0, \ldots, L-1$ be the set of supervised layers at training step $t$; the unnormalized weight is defined as:

$$w_l = \begin{cases} \sum_{i=0}^{l} i, & \text{if } l < L-1 \\ (L-1) + \sum_{i=0}^{L-2} i, & \text{if } l = L-1 \end{cases}\tag{15}$$

These weights are then normalized across $\mathcal{S}$ and used in both the cross-entropy loss and knowledge distillation loss to ensure that deeper exits dominate the overall optimization process, thereby effectively preventing performance degradation at the final output layers.

**Rotational Curriculum Scheduling.** We adopt a rotational curriculum mechanism to activate a subset of exit layers per iteration, ensuring all layers are periodically updated while avoiding conflicting gradients. Let $\mathcal{S}(t)$ be the selected subset of layers at iteration $t$. For a fixed number of partitions $R$ and block size $B = \lceil L/R \rceil$, we define the active block index as:

$$\mathcal{S}(t) = \{ l \mid l \in [(t \bmod R) \cdot B, \ \min((t \bmod R + 1) \cdot B, \ L))] \}\tag{16}$$

This schedule ensures that each layer is supervised once every $R$ steps, enabling more diverse gradient signals across training steps while also improving overall training stability:

$$p_{\ell,t} = S(t) \, D(\ell) \, p_{\max},\tag{17}$$

where $D(\ell)$ controls the per-layer profile and $S(t)$ adjusts the global schedule over training steps.

### A.2 Frequency-Gated Self-Reconstruction: Implementation Details

Given the progressive decoding nature of next-scale generation in VAR, each generation step exhibits distinct spatial resolution and frequency characteristics. To accommodate this, we adopt a **step-wise Frequency-Gated Self-Reconstruction (FGSR)** strategy: a lightweight FGSR module is instantiated for each generation step, allowing frequency-specific supervision that aligns with the evolving spectrum across decoding stages. This setup supports stage-aware, spectrum-adaptive learning while preserving architectural simplicity and inference efficiency.

---

**Algorithm 1** Step-Wise Frequency-Gated Self-Reconstruction Loss

---

1: **Input:** Token maps $\{x_\ell\}$, patch sizes $\{p_i\}$, FGSR modules $\{\text{FGSR}^{(i)}\}$, weight $\lambda$
2: **Output:** Total loss $\mathcal{L}_{\text{FGSR}} + \lambda \cdot \mathcal{L}_{\text{align}}$
3: Initialize $\mathcal{L}_{\text{FGSR}} \leftarrow 0$, $\mathcal{L}_{\text{align}} \leftarrow 0$
4: **for** layer $\ell \in \mathcal{S}$ **do**
5:     Extract $r^\ell \leftarrow x_\ell$
6:     **for** step $i \in \mathcal{A}$ **do**
7:         Extract token slice $r_i^\ell$ of shape $[B, C, p_i, p_i]$
8:         Apply step-specific module: $\hat{r}_i^\ell \leftarrow \text{FGSR}^{(i)}(r_i^\ell)$
9:         $\mathcal{L}_{\text{FGSR}} \mathrel{+}= \|\hat{r}_i^\ell - r_i^\ell\|_2^2$
10:        Extract $W_{\text{fgp}}^{(i)}, W_{\text{inv}}^{(i)}$ from $\text{FGSR}^{(i)}$
11:        $\mathcal{L}_{\text{align}} \mathrel{+}= \|\text{Sym}(W_{\text{fgp}}^{(i)\top} W_{\text{fgp}}^{(i)} - I)\|_F^2 + \|W_{\text{inv}}^{(i)} - W_{\text{fgp}}^{(i)\top}\|_F^2$
12:     **end for**
13: **end for**
14: **return** $\frac{1}{|\mathcal{S}|}\mathcal{L}_{\text{FGSR}} + \lambda \cdot \frac{1}{|\mathcal{S}|\cdot|\mathcal{A}|}\mathcal{L}_{\text{align}}$

---

**Initialization Details.** Each FGSR module operates on the token map $r_t \in \mathbb{R}^{B \times C \times H \times W}$ at step $t$ and applies a wavelet-based frequency decomposition. The learnable exponential gates $\gamma_b$ for each sub-band $b \in \{\text{LL}, \text{LH}, \text{HL}, \text{HH}\}$ are initialized as:

- $\gamma_{\text{LL}} = \log(1)$,
- $\gamma_{\text{LH}} = \gamma_{\text{HL}} = \log(0.5)$,
- $\gamma_{\text{HH}} = \log(1e{-}8)$.

Each FGSR module contains a pair of $1 \times 1$ convolutional layers: one forward projection $W_{\text{fgp}} \in \mathbb{R}^{C \times 4C}$ initialized to be near orthogonal, and one inverse projection $W_{\text{inv}} \in \mathbb{R}^{4C \times C}$ initialized as the transpose of $W_{\text{fgp}}$. These projection layers are shared within the module and are explicitly regularized during training to ensure long-term numerical consistency and stable convergence behavior. We apply FGSR loss to the supervised layers and all decoding steps, using step-specific FGSR modules instantiated for each generation step $i$, as shown in Algorithm 1.

**Adaptive Gating Across Generation Stages.** FGSR enables adaptive frequency gating by responding to the evolving spectral structure across decoding steps. Early token maps primarily capture coarse, low-frequency components, while high-frequency details emerge progressively in later stages. To justify the gating design of FGSR, we provide a view from the information bottleneck principle.

At generation step $t$, the objective is to retain the most relevant information under a distortion constraint. This trade-off can be formulated as a rate-distortion problem:

$$R_t(D) = \min_{p(r_t|x)} I(x; r_t) \quad \text{s.t.} \quad \mathbb{E}[d(x, \hat{x}_t)] \leq D, \tag{18}$$

where $I(x; r_t)$ denotes the mutual information between input $x$ and representation $r_t$, and $d(\cdot)$ is a distortion measure. Let $E_{\text{HF}} = \|\text{HH}\|_2^2$ denote the high-frequency energy. FGSR indirectly controls $E_{\text{HF}}$ via learned sub-band gates $\gamma_b$. From an information-theoretic view, the sensitivity of the rate function to high-frequency energy is given by:

$$\frac{\partial R_t}{\partial E_{\text{HF}}} = \frac{\lambda}{1 + \lambda} \cdot \frac{\|\text{HH}\|_2^2}{\|r_t\|_2^2}, \tag{19}$$

where $\lambda$ is a regularization strength associated with frequency emphasis. This term increases with the decoding steps, which justifies the design of FGSR to progressively activate the high-frequency sub-band gates. Such adaptive behavior ensures early stages prioritize structure and layout, while later stages refine details, maintaining reconstruction fidelity with minimal redundancy. It supports *information-efficient frequency allocation* and aligns with the coarse-to-fine generation paradigm.

### A.3 Gradient-Based Loss Weight Tuning

To achieve a proper balance among the multiple auxiliary losses used during training, such as early-exit supervision, knowledge distillation, and frequency-aware objectives, we perform gradient-based

analysis in the debugging phase. This analysis serves to inform the manual adjustment of the fixed loss weights prior to the final training. At selected training intervals, each loss component $\mathcal{L}_k$ is isolated, and its relative gradient influence with respect to the total loss $\mathcal{L}_{\text{total}}$ is computed as:

$$c_k = \max\left(0, \frac{\langle \nabla\mathcal{L}_{\text{total}}, \nabla\mathcal{L}_k \rangle}{\|\nabla\mathcal{L}_{\text{total}}\|_2^2}\right), \tag{20}$$

where $\nabla\mathcal{L}_k$ and $\nabla\mathcal{L}_{\text{total}}$ denote the flattened gradients with respect to all trainable parameters. The resulting contribution scores $c_k$ are used as a reference to adjust the relative weights of different loss terms to improve the stability and effectiveness of training.

## B  Additional Discussion on High-Frequency Behavior

VAR generates images in a scale-by-scale manner, where coarse-to-fine decoding naturally induces a frequency progression from low-frequency structure to high-frequency details. This behavior follows the Nyquist sampling principle: early steps with low spatial resolution can only represent coarse, low-frequency components, while upsampled later steps allow finer textures to emerge. Moreover, natural images exhibit a $1/f$ power spectrum, implying that most signal energy lies in low frequencies; during training, cross-entropy loss first reduces low-frequency errors, leaving high-frequency residuals to dominate later-stage gradients. This explains the observed "structure-first, detail-later" dynamics in next-scale generation and motivates the frequency-aware supervision strategy used in FreqExit.

## C  Experimental Details

We compare our method **FreqExit** with two representative early-exit baselines, **LayerSkip**[27] and **CoDe**[61]. Since both methods require additional training beyond the original autoregressive model, we detail their training configurations alongside ours below. All methods are trained on the ImageNet-1K dataset using the AdamW optimizer with a global batch size of 1024 and progressive patch scheduling $(1, 2, 3, 4, 5, 6, 8, 10, 13, 16)$.

**CoDe.**  Following the original two-stage pipeline of CoDe, the *drafter* (VAR-d20) is fine-tuned for 15 epochs using cross-entropy loss on the first $N{=}6$ decoding steps, with a learning rate of $1\mathrm{e}{-}6$ and weight decay of $0.08$. The *refiner* (VAR-d16) is then distilled for 65 epochs from the drafter using soft-label supervision. The loss over the first $N$ steps is down-weighted progressively using a dynamic coefficient $\alpha = 1 - \lambda$, where $\lambda$ is the normalized training progress.

**LayerSkip.**  We adapt LayerSkip to our autoregressive model and train it for 80 epochs with a learning rate of $1\mathrm{e}{-}5$ and weight decay of $0.01$. At each step, a subset of transformer layers is supervised in a round-robin fashion with a rotation interval of $R{=}4$, and layer-wise sample dropout is applied with a maximum dropout probability $p_{\max} = 0.1$.

**FreqExit.**  Our method adopts the same optimizer and schedule setup as the other baselines. At each training step, a subset of transformer layers is supervised in a round-robin fashion with group size $R{=}4$ and dropout probability $p_{\max} = 0.1$. Three types of losses are used during training:

- **Early-exit loss:** Applied to selected intermediate layers, this loss combines cross-entropy and distillation terms, each contributing a fixed ratio of 0.5.
- **HF loss:** This wavelet-domain consistency loss is disabled for the first 25% of training, then linearly increased to full strength. Its weight is dynamically computed as

$$\gamma = \alpha \cdot \frac{\bar{\mathcal{L}}_{\text{other}}}{\bar{\mathcal{L}}_{\text{hf}} + \epsilon}, \quad \text{with } \alpha{=}0.3, \tag{21}$$

  where $\bar{\mathcal{L}}_{\text{other}}$ denotes the average of CE, early-exit, and distillation losses.
- **FGSR loss:** This loss promotes frequency-aware representation learning via self-reconstruction. A fixed weight of 5.0 is applied to the FGSR term, along with an orthogonality regularization term weighted by 0.1.

The total loss is computed as the weighted sum of these components. Both HF and FGSR losses are further modulated during training via gradient-based dynamic weight scaling.

Table 4: Training hyperparameters and loss types for methods requiring fine-tuning. BS = batch size, LR = learning rate, WD = weight decay.

| Method | Model | Epochs | BS | LR | WD | $R$ | $p_{\mathbf{max}}$ | Loss Type |
|---|---|---|---|---|---|---|---|---|
| CoDe (drafter) | VAR-d20 | 15 | 1024 | 1e−6 | 0.08 | – | – | CE |
| CoDe (refiner) | VAR-d16 | 65 | 1024 | 1e−5 | 0 | – | – | KD |
| LayerSkip | VAR-d20 | 80 | 1024 | 1e−5 | 0.01 | 4 | 0.1 | CE |
| FreqExit | VAR-d20 | 80 | 1024 | 1e−5 | 0 | 4 | 0.1 | CE + KD, HF, FGSR |

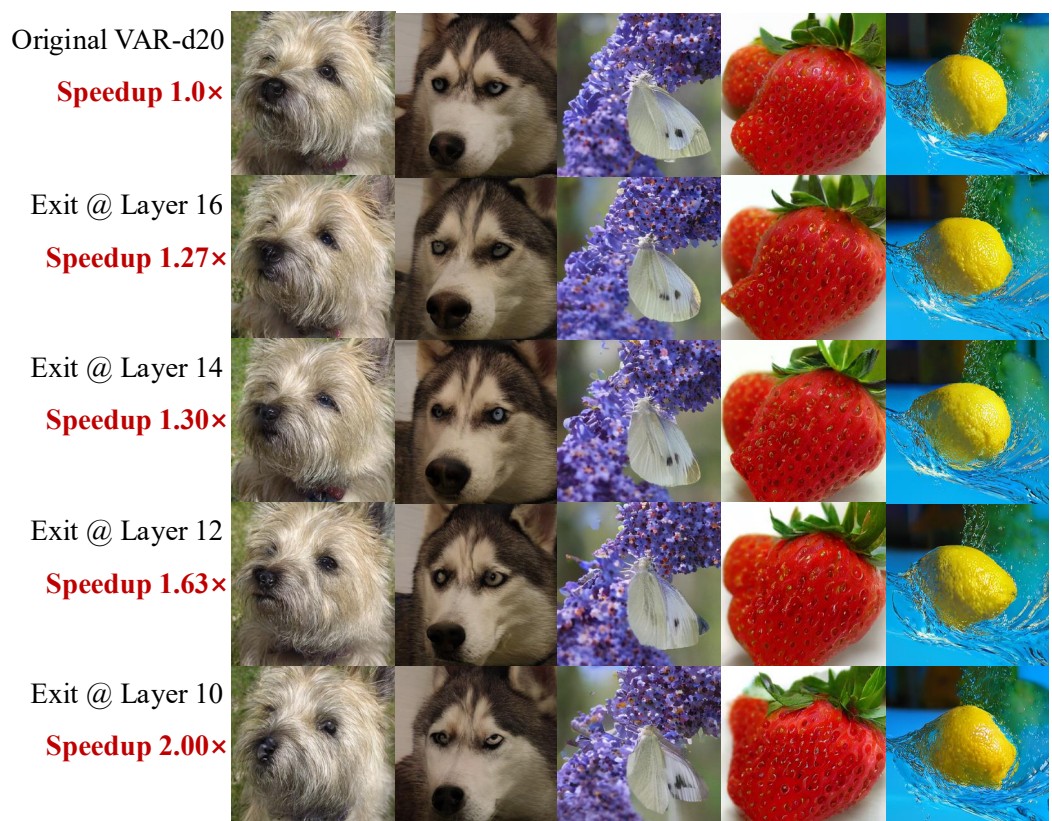

Figure 4: Visualization of generated images from different exit layers under conditional generation. Despite the reduced number of layers, the image quality remains nearly unchanged across exits, while the inference speed can be improved by up to $2\times$.

## D Qualitative Generation Results

**Condition Generation.** We provide an extensive qualitative comparison between the original VAR-d20 model and our proposed FreqExit model fine-tuned with early-exit supervision. As shown in Fig. 4, we visualize outputs from different exit layers (Layer 16, 14, 12, and 10), corresponding to speedups of $1.27\times$, $1.30\times$, $1.63\times$, and $2.00\times$, respectively. From these results, it is evident that even with a two-fold reduction in inference cost, the generated images maintain high visual quality, semantic consistency, and fine-grained detail. This confirms the effectiveness of our approach in enabling efficient autoregressive decoding without compromising fidelity.

**Zero-Shot Inpainting.** We further evaluate the model's capability under zero-shot inpainting settings. As illustrated in Fig. 5, we begin with complete original images and apply random binary masks to simulate missing regions. These masked images, along with class labels, are provided as inputs, and the model performs inpainting using different exit layers. The results demonstrate that our method retains strong generative performance even with early exits. The inpainted regions remain semantically coherent and visually indistinguishable from the ground truth, indicating that the intermediate layers are well-equipped to capture both global structure and local detail. This highlights

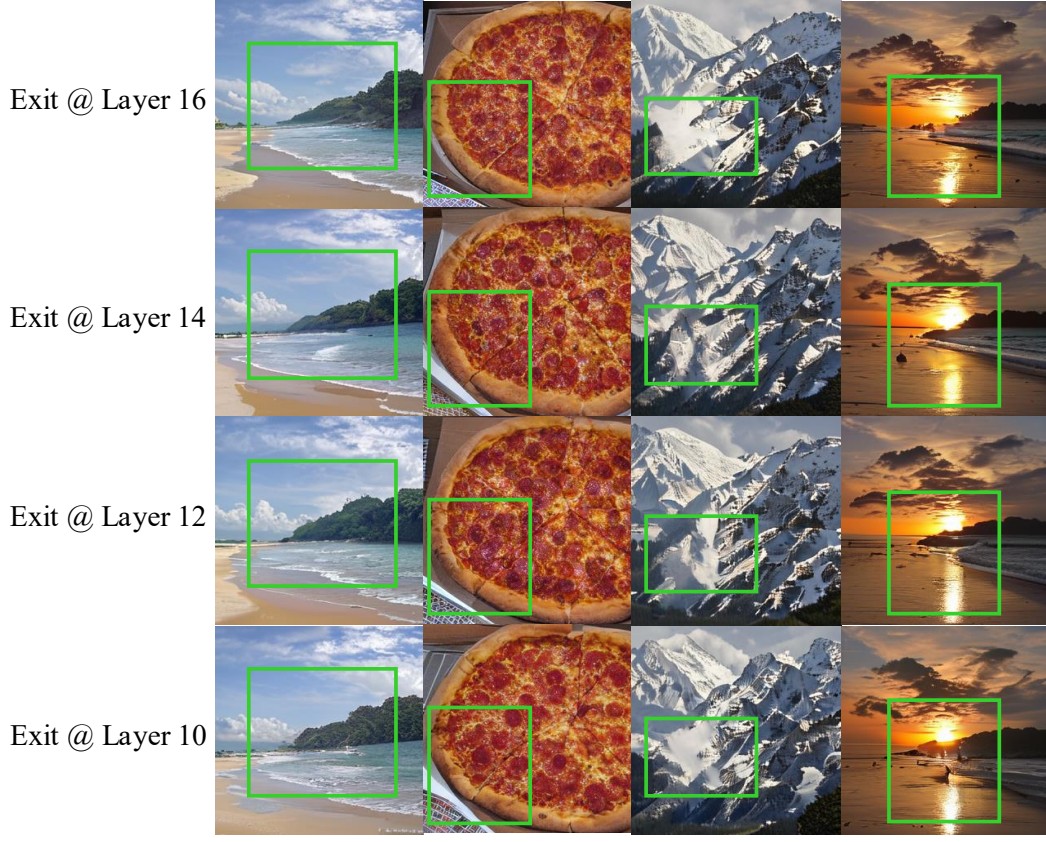

Figure 5: Visualization of zero-shot inpainting results at different exit layers. The model demonstrates strong capability in reconstructing missing regions, and the inpainted content remains visually consistent across different exit depths without introducing artifacts or semantic mismatches.

the capacity of the model for semantic understanding at multiple depths and validates its robustness and adaptability under varying inference conditions.

## E    Discussion and Future Work

We propose **FreqExit**, a unified framework for enabling efficient early-exit inference in next-scale visual autoregressive models such as VAR. By analyzing the spectral dynamics of VAR and addressing the instability of intermediate representations, our method introduces a curriculum-based supervision strategy, a progressive high-frequency consistency loss, and a lightweight frequency-gated self-reconstruction module. These components collectively enhance representation quality and support dynamic exit behavior with minimal impact on generation quality. Experiments on ImageNet 256×256 demonstrate that FreqExit achieves up to **2×** **speedup** with little perceptual degradation, and performs well in both conditional generation and zero-shot inpainting scenarios.

**Limitations.**    While FreqExit demonstrates strong dynamic inference performance, it introduces multiple loss terms to guide intermediate learning. As a result, careful tuning of their weights is needed during training. In particular, we find that the high-frequency consistency loss must be appropriately balanced—if overemphasized, it can hinder the learning of low-frequency structures. Nevertheless, this tuning process is manageable and only needs to be done once per configuration, and a gradient-based analysis used to guide weight selection is provided in Appendix A.

**Future Work**    Future work will explore more adaptive training strategies to better coordinate frequency-aware supervision with semantic objectives, aiming to further improve dynamic inference performance while minimizing the need for manual tuning efforts.

**Reproducibility.** To support reproducibility and further research, we provide training and evaluation code in the supplementary `FreqExit.zip` file and at the anonymous repository `https://github.com/NeuraLiying/FreqExit`.

