# OpenReview forum: "FreqExit: Enabling Early-Exit Inference for Visual Autoregressive Models via Frequency-Aware Guidance"
_NeurIPS.cc/2025/Conference — NeurIPS 2025 poster_

### Official Review · Reviewer_paNB · 2025-06-07

**Clarity:** 2
**Significance:** 2
**Originality:** 2
**Rating:** 4
**Confidence:** 3

**Summary:**

This paper proposes a unified training framework to address the challenges of applying dynamic inference techniques to next-scale autoregressive generation without altering the architecture or compromising output quality. It introduces two components in the curriculum-based training strategy: 1) a high-frequency consistency loss and 2) a lightweight frequency-gated self-reconstruction module, achieving 1.3x speedup without decrease in generation quality.

**Questions:**

1. Can you explain more about what the assumptions of dynamic inference are: semantic stability, monotonic locality, and granularity (line 38); and why these assumptions break in the next-scale generation paradigm? It would strengthen the motivation by explaining the need for a new training strategy for acceleration.
2. How does removing the low-frequency content affect quality? This comparison should be made to support Observation 2 in Section 3.1. Even if this claim is true, why is it important for improving early-exit rather than output quality (line 126)?
3. What is the setting of each strategy in Table 2? Which component enables the early exit? Is it mainly because of the layer dropout?

**Ethical Concerns:**

["NO or VERY MINOR ethics concerns only"]

**Final Justification:**

The authors' justifications make me realize this is a fine-tuning strategy to improve the generation efficiency of VAR model, which is a technique studied before, thus clearing up my initial concern about the need to do so.

The authors' clarifications on the combination of loss functions are valid.

Finally, the authors acknowledge their initially flawed narrative and make a reasonable refinement without over-claiming or causing too much confusion.

Based on the above efforts, I would recommend a borderline accept rating.

**Limitations:**

Yes, the authors include limitations in the supplementary materials. I’m currently concerned about the extensibility to other autoregressive modes beyond the next-scale paradigm, and that the high computational cost may outweigh the improvement in efficiency.

**Paper Formatting Concerns:**

I do not see any major formatting issues.

**Quality:**

2

**Strengths And Weaknesses:**

Strengths:
1. The speedup results are decent: around 30% without quality drop.
2. The observation that the high frequency component matters is quite inspiring.

Weaknesses:
1. Since the proposed acceleration strategy needs to retrain the VAR model from scratch, I am worried that the high computational demand overshadows its gain in efficiency (~30%), and this specifically tailored training strategy may also limit its broader use to other VAR models.
2. The method needs to be evaluated on a broader range of VAR models before it can be claimed as a “unified” training framework.
3. The motivation is not strong enough. The observations made can hardly inform the proposed method. For instance, it’s unclear to me why the three observations, which are made during inference, motivate the need for a new training strategy.
4. It is quite hard to identify which loss function among Eq.(2)(3)(6)(7)(11)(12) contributes the most and may need ablation studies on this. More task-specific insights could improve the logic flow a lot.

---

> ### Author Rebuttal · Authors · 2025-07-31
>
> We sincerely thank you for the recognition and detailed comments. Below, we address each point and clarify the motivation, assumptions, and design choices of our method, hoping to better convey its contributions and resolve your concerns.
>
> **W1: "Since the proposed acceleration strategy needs to retrain the VAR model from scratch ..."**
>
> **W1-Ans:** In practice, **FreqExit does not require retraining the VAR model from scratch**. For example, we start from the official VAR-d20 checkpoint (trained for 250 epochs on ImageNet-1K) and fine-tune for only **80 additional epochs (~30% extra cost)** to enable early-exit capability. Our rotating supervision further reduces computation by selectively activating losses across layers. Thus, the added training overhead is modest and far from the cost of full model training.
>
> This small investment yields sustained inference benefits: ~30% latency reduction without FID loss, and up to 2× speedup with mild quality trade-offs. FreqExit is deployment-friendly and allows runtime adaptation to diverse resource budgets.
>
> Furthermore, FreqExit is the first to apply early exit in next-scale generation, revealing a progressive low-to-high frequency pattern and addressing frequency conflicts via wavelet-based supervision. Our training strategy is general to transformer-style architectures with layered outputs and could extend to autoregressive models in other domains where frequency-phase behaviors are common.
>
> ---
>
> **W2: "The method needs to be evaluated on a broader range of VAR models before it can be claimed as a “unified” training framework."**
>
> **W2-Ans:**  We supplement our analysis with additional experiments on both VAR-d16 and VAR-d24 models. As shown in the tables, our method significantly reduces the FID scores of intermediate layers, demonstrating its effectiveness in enabling early exits with high-quality outputs.
>
> **Table 5: VAR-d16 experimental results**
>
> | Layer   | Epoch 0 | Epoch 16 | Epoch 32 | Epoch 48 |
> |---------|---------|----------|----------|----------|
> | Layer10 | 120.9   | 26.82    | 18.27    | 10.37    |
> | Layer12 | 90.83   | 20.22    | 12.68    | 8.11     |
> | Layer14 | 12.77   | 7.40     | 6.24     | 5.61     |
>
>
> **Table 6: VAR-d24 experimental results**
>
> | Layer   | Epoch 0 | Epoch 4 | Epoch 8  |
> |---------|---------|---------|----------|
> | Layer10 | 134.0   | 68.95   | 27.56    |
> | Layer12 | 129.9   | 60.80   | 17.84    |
> | Layer16 | 127.6   | 41.19   | 8.69     |
> | Layer20 | 85.43   | 21.72   | 5.53     |
>
> ---
>
> **W3: "The motivation is not strong enough."**
>
> **W3-Ans:**
>
> **Table 7: Token Matching Rate with Final Transformer Layer Output (VAR-d20, Last Step)**
> At the last generation step, we apply greedy sampling to each layer’s output and compute the per-token match rate with the final layer’s output tokens.
>
> | Layer | Average Match Rate (%) |
> |-------|------------------------|
> | 14    | 0.51                   |
> | 16    | 0.70                   |
> | 17    | 3.63                   |
> | 18    | 25.2                   |
> | 19    | 59.6                   |
>
> **Motivation:** Our motivation stems from a mismatch between dynamic inference assumptions and next-scale generation behavior in VAR models. CALM-style confidence signals (e.g., Softmax Max, Entropy) remain far below usable thresholds, revealing that training-free heuristics are unreliable. This is because assumptions like semantic stability, monotonic locality, and granularity consistency break down. FreqExit addresses this by introducing training-time supervision to enable reliable early exits.
>
> **Semantic stability:**  assumes that token representations converge across deeper layers. However, as shown in our analysis (Table 1 in Rebuttal ), softmax confidence remains low and unstable throughout the layers of VAR models. This is likely due to the increasing resolution and complexity of token maps, which evolve from coarse global structures to fine-grained high-frequency content. This continual refinement prevents logits from stabilizing and invalidates the assumption that later layers produce redundant outputs.
>
> **Monotonic locality:** assumes that tokens generated at each step can serve as a fixed and verifiable prefix. In next-scale generation, however, token maps are not immutable—tokens generated at step $i$ are frequently updated in step $i{+}1$ through upsampling and refinement. To validate this, we measured the token-wise matching rate between intermediate layers and the final output at the last generation step of VAR-d20. As shown in Table 7, most layers between 10 and 17 yield match rates below 5%, and even Layer 19 reaches only ~60%, confirming the instability of intermediate outputs and the failure of prefix-based verification.
>
> **Granularity consistency:** assumes that dynamic inference operates over fixed, bounded computation units (e.g., one token or one layer). This assumption breaks in next-scale generation due to VAR’s hierarchical structure: each step generates a $p_n \times p_n$ token map, which is upsampled to full resolution and then downsampled for the next step. These tokens are intermediate and overwritten, leading to unstable computation units and growing step-wise cost. As a result, fine-grained control (e.g., per-token or per-layer gating) becomes ineffective.
>
> We believe this unified analysis and supporting experiments clearly demonstrate that effective dynamic inference in next-scale generation requires rethinking beyond inference-time heuristics.
>
> ---
>
> **W4: "It is quite hard to identify which loss function..."**
>
> **W4-Ans:**  Our ablation（Table 8-10）shows that removing either HF-loss or FGSR significantly degrades early-exit quality (higher FID), while final-layer outputs are only marginally affected—supporting our claim that high-frequency mismatch is the main constraint on early-exit performance.
>
> **Table 8: VAR-d20 with Early Exit loss**
> | Layer   | Epoch 40 | Epoch 48 | Epoch 56 | Epoch 64 |
> |---------|----------|----------|----------|----------|
> | Layer 8 | 22.73    | 20.53    | 19.40    | 18.52    |
> | Layer 12| 6.87     | 5.52     | 5.31     | 5.21     |
> | Layer 16| 3.64     | 3.55     | 3.45     | 3.42     |
>
> **Table 9: VAR-d20 with Early Exit loss and HF loss**
> | Layer    | Epoch 40 | Epoch 48 | Epoch 56 | Epoch 64 |
> |----------|----------|----------|----------|----------|
> | Layer 8  | 16.22    | 15.43    | 14.67    | 13.92    |
> | Layer 12 | 4.83     | 5.32     | 4.93     | 4.72     |
> | Layer 16 | 3.41     | 3.38     | 3.10     | 3.06     |
>
> **Table 10: VAR-d20 with Early Exit loss, HF loss, and FGSR loss (FreqExit)**
> | Layer    | Epoch 40 | Epoch 48 | Epoch 56 | Epoch 64 |
> |----------|----------|----------|----------|----------|
> | Layer 8  | 13.20    | 12.54    | 11.74    | 10.90    |
> | Layer 12 | 4.59     | 4.46     | 4.37     | 3.92     |
> | Layer 16 | 3.27     | 3.06     | 2.91     | 2.87     |
>
> ---
>
> **Q1: "Can you explain more about what the assumptions of dynamic inference are: semantic stability, monotonic locality, and granularity..."**
>
> **Q1-Ans:**  VAR models exhibit unstable intermediate tokens, overwritten outputs across steps, and variable computation granularity, undermining conventional early-exit heuristics. These failures motivate our training-time solution. Please refer to W3-Ans for full analysis and evidence.
>
> ---
>
> **Q2: "How does removing the low-frequency content affect quality?"**
>
> **Q2-Ans:**  We clarify the roles of low- and high-frequency components in image generation: low-frequency (LL) sub-bands capture global semantics, while high-frequency (LH/HL/HH) sub-bands encode fine textures. Evaluating VAR-d16/d20/d24/d30, we found that removing low-frequency components leads to **FID > 200**,  indicating **major semantic distortion**, whereas removing high-frequency components mainly affects perceptual quality (Fig.~2(b)). This confirms that low-frequency governs semantics, while high-frequency governs realism.
>
> As shown in Fig.~2(a), high-frequency energy rises steadily during generation, reaching ~50% in later stages. This makes high-frequency mismatch a key source of residual error, as shallow layers struggle to capture the needed fine details. In such cases, enhancing low-frequency features helps little, while improving high-frequency modeling yields significant FID gains. FreqExit introduces two high-frequency enhancement strategies:
>
> **(1) HF Consistency Loss:** aligns the wavelet-domain high-frequency outputs of shallow layers with the final output.
>
> **(2) FGSR Loss:** provides stage-aware, spectrum-adaptive supervision, guiding the model to adjust to frequency behavior at each generation step and improve texture reconstruction.
>
> Our ablation（Table 8-10）shows that removing either HF-loss or FGSR significantly degrades early-exit quality (higher FID), while final-layer outputs are only marginally affected. In conclusion, Observation 2 highlights a critical insight: once low-frequency information is learned through early-exit supervision, remaining quality gaps are due to insufficient high-frequency modeling.
>
>
> ---
>
> **Q3: "What is the setting of each strategy in Table 2?"**
>
> **Q3-Ans:**  The strategies in Table 2 represent different early-exit paths applied to the same trained FreqExit model. Each is a 10-dimensional vector specifying the number of layers used at each generation step. For example,
> “7” = [16, 12, 10, …, 10] uses 16 layers at the first step, 12 at the second, and 10 for the rest.
> This enables flexible quality–latency trade-offs without retraining or extra modules.
>
> This early-exit capability is not achieved by Layer Dropout. It is enabled by our training design, which equips each layer with independent generation ability via supervised learning. Layer Dropout is only used during training as a regularizer to improve robustness and has no effect at inference time.

---

> ### Author Response · Authors · 2025-08-04
> **Expanded Evaluation on Next-Token VAR and Training Objective Clarification**
>
> We appreciate the reviewer’s thoughtful follow-up comments and address each point in detail below.
>
> ---
> **Q1: "...but VAR is more than just next-scale, like next-token. To me, this cannot be claimed as “unified” unless the authors can justify its applicability to other VAR paradigms."**
>
> **Ans-1:** Thanks for the suggestion! To validate the applicability of our method to **next-token** generation VAR models, we add the experiments on LlamaGen [^1].  We finetune LlamaGen-B (∼111M parameters, 12 hidden layers) on the ImageNet dataset using the proposed FreqExit method and evaluate the FID of intermediate-layer outputs. Results are shown below.
>
>  As shown in Table 1, the FID scores consistently decrease across epochs for layers 8–11, suggesting that FreqExit is also applicable to next-token generation models.
>
> **Table 1: FID scores of intermediate-layer outputs for FreqExit on LlamaGen-B.**
> | Layer | Origin | Epoch 4 | Epoch 8 | Epoch 12 | Epoch 16 | Epoch 20 |
> |-------|--------|---------|---------|----------|----------|----------|
> | 8     | 202    | 57.4    | 47.9    | 39.0     | 37.9     | 36.2     |
> | 9     | 176    | 35.1    | 28.6    | 25.4     | 22.8     | 22.4     |
> | 10    | 44.3   | 26.9    | 19.7    | 17.2     | 16.3     | 14.4     |
> | 11    | 11.1   | 11.3    | 11.2    | 11.1     | 10.9     | 10.8     |
>
> Besides, additional experiments are conducted on the larger LlamaGen-L model (∼343M parameters, 24 hidden layers). For comparison with the baseline used in the main paper, we also fine-tune LlamaGen-L using the LayerSkip method, which applies only cross-entropy loss without frequency-aware supervision. Both models are fine-tuned for 20 epochs, and the FID results are shown in Table 2 (FreqExit) and Table 3 (LayerSkip).  Table 2 demonstrates that FreqExit remains effective on a larger next-token generation model. Compared to LayerSkip (Table 3), **FreqExit achieves lower FID scores and faster convergence.** These results validate the applicability of our method beyond next-scale VAR.
>
> **Table 2: FID scores of intermediate-layer outputs for FreqExit on LlamaGen-L.**
> | Layer | Origin | Epoch 4  | Epoch 8  | Epoch 12 | Epoch 16 | Epoch 20 |
> |-------|--------|------|------|------|------|------|
> | 20    | 256    | 36.3 | 11.3 | 10.3 | 10.0 | 9.9  |
> | 21    | 207    | 30.5 | 8.9  | 8.6  | 8.5  | 8.2  |
> | 22    | 59.2   | 20.0 | 8.9  | 8.5  | 8.2  | 7.8  |
> | 23    | 37.7   | 18.9 | 8.0  | 7.5  | 7.5  | 7.4  |
>
>
> **Table 3: FID scores of intermediate-layer outputs for LayerSkip on LlamaGen-L.**
> | Layer | Epoch 4  | Epoch 8  | Epoch 12 | Epoch 16 | Epoch 20 |
> |-------|------|------|------|------|------|
> | 20    | 52.7 | 22.0 | 16.4 | 13.4 | 11.9 |
> | 21    | 40.0 | 12.1 | 10.9 | 10.2 | 9.5  |
> | 22    | 34.3 | 9.6  | 9.2  | 8.7  | 8.8  |
> | 23    | 22.8 | 9.3  | 9.1  | 8.8  | 8.5  |
>
>
> References:
> - [1] Autoregressive Model Beats Diffusion: Llama for Scalable Image Generation, Arxiv: 2406.06525
>
> ---
>
> **Q2: "Early Exit loss (Eq.(2)), HF loss (Eq.(6)), and FGSR loss (Eq.(11)) are used as the loss functions? It is unclear how the distillation loss in Eq.(3) and the alignment regularizer in Eq.(12) function differently. It would be good to present the final training objective, which is currently lacking in the paper."**
>
> **Ans-2:** We appreciate the suggestion to clarify the overall training objective.  As requested, the final training loss used for FreqExit is defined as:
>
> $$
> L_{\text{total}} = \lambda_1 L_{\text{CE}} + \lambda_2 L_{\text{KD}} + \lambda_3 L_{\text{HF}} + \lambda_4 L_{\text{FGSR}}
> $$
>
>
> - *$L_{\text{CE}}$* and *$L_{\text{KD}}$* (Eq.(2) and Eq.(3)) together constitute the **early-exit supervision loss**, which provides the primary learning signal by aligning intermediate predictions with ground-truth labels and final-layer outputs.
> - *$L_{\text{HF}}$* and *$L_{\text{FGSR}}$* (Eq.(6) and Eq.(11)) serve as the **frequency adaptation loss**, which further enhances the high-frequency reconstruction capability of early exits and improves their perceptual quality.
>
> Note that $L_{\text{FGSR}}$ includes an alignment regularizer $L_{\text{align}}$ (Eq.(12)), which enforces approximate orthogonality in the projection matrices to stabilize spectrum reconstruction. Since it is an internal component of $L_{\text{FGSR}}$, we do not treat it as a separate loss term.
>
> ---
>
> We hope the additional results and clarifications can fully resolve your concerns. If you have extra critical concerns, we would be grateful if you could let us know early so that we have enough time to address them.
>
> *Finally, thank you so much for helping us improve the paper so far! Looking forward to your further feedback.*

---

> > ### Author Response · Authors · 2025-08-06
> > **Kindly requesting feedback from Reviewer paNB**
> >
> > Dear Reviewer paNB,
> >
> > Thank you once again for your valuable comments on our submission! We have posted responses to your concerns and included additional analyses and experiments, which we believe have adequately addressed your concerns.
> >
> > We totally understand that this is quite a busy period, so we deeply appreciate it if you could take some time to return further feedback on whether our response solves your concerns. If there are any other comments, we will do our best to address them in the rest of the author-reviewer discussion period. Thank you very much!
> >
> > Sincerely,
> >
> > Authors

---

> > > ### Comment · Reviewer_paNB · 2025-08-08
> > >
> > > Thank you for the additional experiments and clarification.
> > >
> > > Based on the authors’ justification, the proposed method appears to be a general-purpose strategy to improve inference for VAR. However, the paper initially claims it is specifically tailored to the next-scale paradigm, and the analysis (e.g., Figure 3(b)) also seems focused on this setting.
> > >
> > > The authors should clarify their positioning: either present the method as a targeted improvement for next-scale VAR and remove the “unified” claim, or emphasize its general applicability and drop the “specifically tailored” description. **Doing both creates a confusing and inconsistent narrative.** I encourage the authors to reconsider this point and revise the writing to support a coherent and well-founded claim.
> > >
> > > This issue is my primary concern to change the rating.

---

> > > > ### Author Response · Authors · 2025-08-08
> > > > **Clarification of “unified” and positioning**
> > > >
> > > > We sincerely thank the reviewer for the valuable feedback and for pointing out the potential ambiguity in our positioning. We fully understand the concern about maintaining a coherent and consistent narrative, and we address it in detail below.
> > > >
> > > > **(1) Explanation of the “unified” in the original paper**
> > > >
> > > > In the **original paper**, the term *“unified”* refers to our **integrated loss design**, not to the applicability of the method across all VAR paradigms. As stated in Section 3.2, *“we propose a unified training framework … integrating curriculum-based supervision, a frequency-aware consistency loss, and the FGSR module.”*  Here, the “*unified”* **was intended to describe** the joint design of multiple complementary losses and modules within the **training objective**, rather than a claim of universal applicability.
> > > >
> > > > We did not expect the term “unified” was interpreted differently by the reviewer and caused the confusion, but this indeed showed our writing was not clear enough - great thanks to the reviewer for raising this up and thus we had the opportunity to eliminate this confusion.
> > > >
> > > > **(2) Additional experiments and possible extension**
> > > >
> > > > Following the reviewer’s suggestion, we have conducted experiments on a next-token generation VAR model (LlamaGen). The results show that FreqExit remains effective, suggesting potential applicability beyond the next-scale setting, although this is not our original design motivation.
> > > >
> > > > **(3) Revisions for coherence**
> > > >
> > > > We agree with the comment on the need for a coherent and consistent narrative. To address this, we will:
> > > >
> > > > - **Remove the *“unified”* claim** to avoid misinterpretation, and replace it with *“**integrated loss design**”* to more precisely describe our contribution.
> > > > - Clearly position FreqExit as a method **specifically designed for next-scale VAR**.
> > > > - We considered trying to claim more generally. But frankly, we feel the current experiments on the next-token VAR (LlamaGen) are insufficient for us to claim so responsibly. Therefore, we decide to stick to the original scope of the paper and present next-token VAR results only as **supplementary evidence of the very promising extension**, without over-claiming.
> > > >
> > > > ---
> > > >
> > > > We sincerely thank the reviewer again for the valuable feedback and constructive suggestions. We believe these clarifications and revisions have adequately addressed your concerns. We would be grateful if you could kindly adjust the score in light of these improvements. Let us know if you have follow-up questions. Thanks!

---

> ### Comment · Reviewer_paNB · 2025-08-08
>
> Thank you for the response. That clears up my concerns. I will raise the score, assuming the authors will make adjustments to the writing and integrate the answer to my previous questions in the revised version accordingly.

---

> > ### Author Response · Authors · 2025-08-09
> >
> > Dear Reviewer paNB,
> >
> > Thank you for kindly raising the score!
> >
> > In the revised version, we will refine the writing for clarity and add the discussions and experimental results from the rebuttal.
> >
> > Sincerely,
> >
> > Authors

---

### Official Review · Reviewer_sjr1 · 2025-06-26

**Clarity:** 2
**Significance:** 2
**Originality:** 3
**Rating:** 4
**Confidence:** 4

**Summary:**

This paper proposes a dynamic inference framework tailored for the next-scale prediction (VAR) paradigm by integrating an early-exit mechanism. The approach achieves up to 2× speedup with minimal performance degradation.

**Questions:**

See weaknesses, more concise and clear explanations and comprehensive analysis are needed.

**Ethical Concerns:**

["NO or VERY MINOR ethics concerns only"]

**Final Justification:**

This work conducts thorough experiments on each component and systematically compares with existing baselines, clearly demonstrating the superiority of the proposed method. I believe it is a valuable contribution to the community.

**Limitations:**

yes

**Quality:**

3

**Strengths And Weaknesses:**

Strengths:

The paper is clearly written and well-structured. It provides a thorough analysis of the frequency-domain properties and intermediate feature distributions in VAR, and designs a fine-tuning strategy for early-exit accordingly. Compared to prior acceleration methods for VAR, this approach achieves better performance under higher speed-up ratios.

Weaknesses:

1. In Equation (2), the meaning of $Y$ is not explained. The term $\tilde{e}$ seems to be a dynamic weight, but a more detailed explanation is needed. Moreover, is the total loss simply a summation of all the losses proposed in the Method section?
2. Although many losses are introduced in the Method section, the ablation study only investigates the HF and FGSR losses. It would be helpful to see how other components affect performance—for instance, the role of the $\tilde{e}$ and gate terms in the early-exit loss, and the temperature scheduling in Equation (3).
3. The overall framework feels a bit complex, making it hard to pinpoint which parts contribute most to the performance. A more concise and clearer explanation would greatly improve readability.
4. If possible, please provide some visual comparison with the baseline model (e.g., CoDe and LayerSkip).

---

> ### Author Rebuttal · Authors · 2025-07-31
>
> We sincerely thank the reviewer for the thoughtful and constructive feedback. Below, we provide point-by-point responses to the comments and questions.
>
> ---
> **W1: "In Equation (2), the meaning of $Y$ is not explained. The term $\tilde{e}$ seems to be a dynamic weight, but a more detailed explanation is needed. Moreover, is the total loss simply a summation of all the losses proposed in the Method section?"**
>
> **W1-Ans**: We thank the reviewer for pointing this out. We provide the following clarifications and will annotate the definitions in the revised version:
>
> - Definition of $Y$: $Y$ denotes the ground-truth token sequence obtained by encoding the target image with a pre-trained VQ-VAE, which serves as the supervision target for predictions.
>
> - Definition of $\tilde{e}(t,\ell)$: $\tilde{e}(t,\ell)$ follows the formulation in Eq. (2), and is a normalized dynamic weight that determines the contribution of each intermediate layer $\ell$ to the early-exit loss at training step $t$:
>
> where
>
> - $C(t,\ell)$ is a curriculum gate that selects a subset of layers to be supervised at each training step. Specifically, all layers are divided into $R$ blocks (with $R=4$ for VAR-d20), and one block is activated per step in a round-robin schedule: block_id = $t$ mod $R$.
>
> - $e(\ell)$ is a depth-dependent static weight defined as: (1) If $\ell < L-1$ , $e(\ell) =     \sum_{i=0}^\ell i$, (2) if $\ell= L-1$ , $e(\ell) =  (L{-}1) + \sum_{i=0}^{L{-}2} i $. This formulation ensures that deeper layers receive quadratically increasing weights, with additional emphasis on the final layer to promote more comprehensive supervision. This design corresponds to Eq. (14) in Appendix A.1.
>
> - The normalization ensures that the sum of $\tilde{e}(t,\ell)$ over all active layers is 1. This design emphasizes deeper layers during training, ensuring that updates to shallow layers do not compromise the performance of deeper ones.
>
> **Total loss structure:** The total loss is a weighted combination of the main task loss (CE + KD), the high-frequency (HF) loss, and the FGSR loss. The main loss is applied to both the final and intermediate layers using $\tilde{e}(t,\ell)$. The HF loss is activated after 25% of training and dynamically scaled (see Appendix B, Eq. (20)), while the FGSR loss is applied with a fixed weight of 5.0, along with an orthogonality term weighted by 0.1.
>
> Details on the layer-wise loss weights and training schedule can be found in Appendix B (FreqExit). We hope this clarifies the formulation and thank the reviewer again for the valuable feedback.
>
> ---
>
> **W2: "Although many losses are introduced in the Method section, the ablation study only investigates the HF and FGSR losses. It would be helpful to see how other components affect performance—for instance, the role of the $\tilde{e}$ and gate terms in the early-exit loss, and the temperature scheduling in Equation (3)."**
>
> **W2-Ans**: The roles of the dynamic weighting $\tilde{e}(t,\ell)$ and the gating mechanism $C(t,\ell)$ have been detailed in our response to Comment 1. Briefly, $e(\ell)$ emphasizes deeper layers to protect their optimization from shallow-layer interference, while $C(t,\ell)$ reduces per-step computational cost and gradient conflict by supervising only a subset of layers in a rotational schedule. Together, they help stabilize training and enhance early-exit performance.
>
> We also provide further clarification on temperature scheduling in Eq. (3). The temperature $T_\ell$ decreases linearly from 4.0 at shallow layers to 1.0 at the final layer.  This design is based on the observation that shallow layers have limited semantic capacity during early training. A higher temperature smooths the teacher distribution, providing softer and more tolerant supervision targets. As depth increases, semantic representations become more refined; hence, lower temperatures help retain sharper predictions and improve alignment. This strategy aligns with prior work on progressive distillation and improves intermediate representation quality.
>
> ---
>
> **W3: "The overall framework feels a bit complex, making it hard to pinpoint which parts contribute most to the performance. A more concise and clearer explanation would greatly improve readability."**
>
> **W3-Ans**: We thank the reviewer for the helpful suggestion. To clarify the contribution of individual components, we conducted additional ablation studies to further isolate the effects of the FGSR module and temperature scheduling.
>
>
> ----
>
> **W4: "If possible, please provide some visual comparison with the baseline model (e.g., CoDe and LayerSkip)."**
>
> **W4-Ans**: We thank the reviewer for the helpful suggestion. We agree that visual comparisons with baseline methods would provide valuable qualitative insight. Due to rebuttal constraints, we are unable to include figures at this stage. However, we will incorporate representative visual comparisons in the revised version of the paper to highlight the differences in generation quality.

---

> > ### Comment · Reviewer_sjr1 · 2025-08-04
> >
> > I have carefully reviewed the authors' responses to my comments, which have addressed some of my concerns. I  will keep my score, and hope the authors can further improve these aspects in the next revision by providing clearer quantitative/qualitative comparisons and more comprehensive ablation studies.

---

> ### Author Response · Authors · 2025-08-06
> **Additional ablation results and analysis are provided for the layer weighting and temperature scheduling components.**
>
> Thank you for your feedback. Due to the rebuttal deadline extension, we have extra time to include additional quantitative experiments to address your concerns.  we add new ablation studies on the effects of the layer weighting scheme $\tilde{e}$ and the temperature scheduling strategy. The experiments are conducted on the LlamaGen-L model (343M, 24 layers). Table 1 reports the results comparing the uniform-weight baseline ($\tilde{e}=1$) with our depth-aware layer weighting scheme. Table 2 compares the effect of using a fixed temperature ($T=1$) versus our progressive scheduling strategy, where $T$ decreases linearly from 4.0 at shallow layers to 1.0 at the final layer.
>
> ---
>
> **Table 1. FID scores of intermediate-layer outputs under two layer weighting settings: (1) uniform weighting ($\tilde{e}=1$), and (2) depth-aware weighting computed according to Eq.~(14) in the paper. Results are reported at epochs 8, 16, and 20 on LlamaGen-L.**
>
> | **Layer** | **ep8  ($\tilde{e}=1$)** | **ep16  ($\tilde{e}=1$)** | **ep20 ($\tilde{e}=1$)** | **ep8 (ours)** | **ep16 (ours)** | **ep20 (ours)** |
> | --- | --- | --- | --- | --- | --- | --- |
> | 20 | 10.8 | 9.9 | 9.9 | 11.3 | 10.0 | 9.9 |
> | 21 | 8.8 | 8.4 | 8.4 | 8.9 | 8.5 | 8.3 |
> | 22 | 9.2 | 8.3 | 8.3 | 8.9 | 8.2 | 7.8 |
> | 23 | 12.3 | 8.5 | 8.6 | 8.0 | 7.5 | 7.4 |
>
> ---
>
> **Table 2. FID scores of intermediate-layer outputs under two temperature settings: (1) fixed temperature $T=1$, and (2) progressive temperature scheduling from $T=4.0$ (shallow layers) to $T=1.0$ (final layer). Results are reported at epochs 8, 16, and 20 on LlamaGen-L.**
>
> | **Layer** | **ep8 ($T=1$)** | **ep16 ($T=1$)** | **ep20 ($T=1$)** | **ep8 (ours)** | **ep16 (ours)** | **ep20 (ours)** |
> | --- | --- | --- | --- | --- | --- | --- |
> | 20 | 14.8 | 11.0 | 10.6 | 11.3 | 10.0 | 9.9 |
> | 21 | 9.8 | 9.1 | 8.8 | 8.9 | 8.5 | 8.3 |
> | 22 | 9.2 | 8.7 | 8.3 | 8.9 | 8.2 | 7.8 |
> | 23 | 8.4 | 8.5 | 8.3 | 8.0 | 7.5 | 7.4 |
>
> ---
>
> - **Layer weighting**
>
> Our model involves multi-layer supervision, which can be seen as a multi-objective optimization problem. To prevent shallow-layer gradients from dominating and degrading deep-layer performance, we assign progressively increasing weights to deeper layers. As shown in Table 1, using uniform weights ($\tilde{e}=1$) yields similar FID at shallow layers (e.g., layer 20). However, it leads to worse performance at deeper layers (e.g., layers 22–23), indicating interference from shallow-layer updates.
>
> ---
>
> - **Temperature scheduling**
>
> In early-exit settings, aligning shallow-layer outputs with the final-layer teacher logits can be overly strict due to limited representational capacity. Applying a higher temperature smooths the teacher distribution, reducing the difficulty of distillation for shallow layers. As shown in Table 2, progressive temperature scheduling reduces FID across all layers, with the effect especially pronounced at deeper ones. This reduction is likely related to more stable alignment at intermediate layers.
>
> ---
>
> We hope these additional analyses help clarify our design choices and address your concerns. If you find the improvements satisfactory, we would be grateful if you would kindly consider raising the score.
>
> If you have any further suggestions or questions, we would be happy to hear them and will do our best to respond within the available time. Thank you again for your thoughtful feedback and we truly appreciate your time and consideration.

---

> > ### Comment · Reviewer_sjr1 · 2025-08-07
> >
> > Thank you for the additional experimental results. The authors have demonstrated that the proposed layer-wise weighting scheme and progressive temperature adjustment play a significant role in improving training performance, and the accompanying explanations are reasonable.
> > My concerns have been addressed. Overall, this is a solid paper, and I hope the authors incorporate these improvements in the final version.

---

### Official Review · Reviewer_NPSH · 2025-07-03

**Clarity:** 3
**Significance:** 3
**Originality:** 3
**Rating:** 4
**Confidence:** 3

**Summary:**

FreqExit proposes a unified training framework to accelerate Visual AutoRegressive Modeling through frequency-aware supervision. They observe that high-frequency details, which are important for image perception, appear in the later stages of the generation, and there is an instability in the intermediate representations. To solve these problems and encourage early exit, they use a layer and time-dependent dropout, early exit loss, and a curriculum scheduling. They also introduce a frequency-gated residual connection to learn high-frequency details at earlier stages without affecting the main architecture. Through empirical analysis, they show that they are able to achieve almost 2x speed up in image generation with minimal degradation in quality.

**Questions:**

1. Other datasets and different resolutions can provide more strength to the generalizability of the methods used in the paper
2. Though the appearance of high-frequency terms in the later stages of generation is observed empirically, the reasons behind it are not made clear in the paper
3. The details of the wavelet transform used is not specified in the paper. What about different frequency analysis like the frequency or cosine transforms?

**Ethical Concerns:**

["NO or VERY MINOR ethics concerns only"]

**Final Justification:**

Some of my concerns were addressed in the rebuttal by the authors; I am keeping my favorable score.

**Limitations:**

yes

**Paper Formatting Concerns:**

Minor typos:  "for for" in line 22, "allowingg" in line 253.
Figure 3 is cutting off the title of section 3.1

**Quality:**

3

**Strengths And Weaknesses:**

Strengths:
1. The authors introduce a suite of techniques to accelerate image generation in VAR modeling, tackling multiple challenges
2. The observation frequency characteristics developing over time are new and helped guide their methodology for FGSR and high-frequency consistency loss
3. Curriculum-based early exit supervision encourages the earlier layers of the model to learn robust representations in the early layers, thereby enabling speed-up
4. Post training, their framework allows dynamic early-exit strategies trading off generation quality and speedup.

Weaknesses:
1. Their hypothesis that high frequency terms improve perception quality runs counter to the idea of natural image compression techniques, such as JPEG, that prune high frequency terms while keeping low frequency terms
2. Their distilling process needs two models, which increases training costs for large-scale models
3. All the experiments are shown on ImageNet generation, lacking diversity in the evaluations.

---

> ### Author Rebuttal · Authors · 2025-07-30
>
> We sincerely thank the reviewer for the thoughtful and constructive feedback. Below, we provide point-by-point responses to the comments and questions.
>
> ---
>
> **W1: Role of High Frequencies**
>
> **W1-Ans:** We thank the reviewer for raising this point. To clarify upfront: image compression (e.g., JPEG) and visual image generation **pursue fundamentally different goals**. Compression removes redundant information from complete images to reduce bitrate. In contrast, generative models synthesize content from scratch and **rely on high-frequency details** to achieve perceptually convincing results. In generative methods such as GANs [1] and diffusion models [2], insufficient high-frequency modeling typically leads to over-smoothing and loss of texture, degrading visual quality.
>
> Our work investigates this issue in autoregressive visual generation, with a focus on **next-scale generation**, where images are produced progressively across resolution scales. We identify a **previously underexplored frequency pattern**: low-frequency components dominate early stages, while high-frequency content becomes dominant in later steps. This frequency inconsistency limits the effectiveness of confidence-based, training-free early-exit strategies, which assume uniform information distribution across layers. To address this, we propose FreqExit, a frequency-adaptive training framework that explicitly aligns learning dynamics with this progression. It integrates frequency-aware supervision and a frequency-gated residual mechanism to promote earlier modeling of high-frequency features, enabling more effective early exits with minimal loss in image quality.
>
> ---
>
> **W2: Distillation Overhead**
>
> **W2-Ans:** In our setup, the teacher model is frozen and performs only a single forward pass per batch. Its activations are not stored, and memory is immediately released after inference, resulting in minimal resource usage. To quantify this, we conducted a profiling experiment using PyTorch’s `torch.cuda.max_memory_allocated()` to capture peak memory during the teacher’s forward pass. As shown in Table 4, the teacher model accounts for only **5.3%** of peak memory usage and **8.8%** of total training time.
>
> **Table 4: Resource Usage Breakdown During Distillation Training**
>
> | Component                      | Peak Memory Usage (%) | Time Usage (%) |
> |-------------------------------|------------------------|----------------|
> | Teacher Model (Forward Only)  | 5.3%                   | 8.8%           |
> | Student Model (Training Total)| 94.7%                  | 91.2%          |
>
>
> This confirms that our distillation strategy imposes **minimal overhead**. The training process is also fully compatible with widely-used optimization tools such as mixed-precision training, ZeRO, and DeepSpeed, which can further reduce the cost if necessary. The modest additional training cost is acceptable, and this frozen-teacher strategy is widely adopted in model compression research [3-4].
>
> ---
>
> **W3: Dataset and Resolution Generalization**
>
> **W3-Ans:**  We chose ImageNet-256 as our evaluation benchmark because existing VAR-style models are trained and publicly released on this dataset. This ensures reproducibility and allows for direct, fair comparison with baseline methods. In addition, ImageNet’s 1,000-category diversity provides rich variations in texture, scale, and semantics, making it a strong testbed for evaluating convergence, detail recovery, and long-range dependencies.
>
> FreqExit is **resolution- and dataset-agnostic**, as it operates solely along the depth axis by adjusting the number of active layers at each generation step, without modifying the patching scheme, tokenization process, or data modality. For example, in VAR models, increasing resolution from 256×256 to 512×512 enlarges the token maps at each step (e.g., step-wise `patch_nums` from (1, 2, …, 16) to (1, 2, …, 32)), but the per-step computation graph remains unchanged, allowing FreqExit to scale without modification. Recent work such as Infinity [5] has already extended VAR-style architectures to 1024+ resolutions while preserving their core structure, indirectly supporting the transferability of our method.
>
> We agree that broader evaluation across datasets and resolutions would further demonstrate generalizability. We plan to extend FreqExit in these directions, but due to time and resource constraints, we are unable to provide new experiments during the rebuttal phase and appreciate the reviewer’s understanding.
>
> ---
>
> **Q1: Generalization of FreqExit**
>
> **Q1-Ans:**   All publicly available VAR baselines are trained on ImageNet at 256 × 256, so we used the same setting for reproducible, fair comparison. FreqExit itself is resolution- and dataset-agnostic because it only adjusts the active‐layer depth. Training higher-resolution models (e.g., 512 × 512) would take about 24 days on 8 × A100 GPUs, which is infeasible during the rebuttal period. We therefore defer larger-scale and cross-dataset experiments to future work.
>
> ---
>
> **Q2: Discussion on High-Frequency Behavior**
>
> **Q2-Ans:** VAR generates images in a scale-by-scale manner, where each step operates on a low-resolution token map that is upsampled before the next step. In early steps, the coarse spatial resolution limits the range of frequency components that can be represented—only large-scale structures and smooth regions (i.e., low-frequency information) can be reliably synthesized. This limitation aligns with the Nyquist sampling principle, which states that **higher-frequency content requires denser spatial sampling**. As the token map is upsampled, the spatial granularity increases, allowing later layers to generate finer textures and high-frequency details. This leads to a natural “structure-first, detail-later” generation process.
>
> From an information-theoretic perspective, natural images typically follow a $1/f$ power spectrum, where most of the signal energy is concentrated in the low-frequency range [6]. During training, cross-entropy loss tends to reduce large-magnitude errors first—primarily from low-frequency components. As structural errors diminish, high-frequency residuals become more prominent in the gradient signal, which explains the gradual emergence of fine details in later stages.
>
> ---
>
> **Q3: Wavelet Details**
>
> **Q3-Ans:** We use a single-level 2D Haar wavelet transform (`wave=‘haar’`) in FreqExit. Each $2\times2$ spatial block in the token map is linearly projected into four subbands: LL (low-frequency structure) and LH/HL/HH (directional high-frequency details). This fixed-stride decomposition preserves a one-to-one spatial correspondence between inputs and outputs, allowing frequency-aware gating and reconstruction in a unified coordinate system without extra alignment.
>
> Compared to global Fourier transforms or block-wise DCT, Haar wavelets are better suited to our setting for three main reasons:
>
>  (1) **Spatial-frequency alignment**: FFT and DCT distribute frequency information globally or within local blocks, weakening spatial localization. In contrast, Haar wavelets retain both **spatial and frequency information**, which is better suited for frequency-aware control in spatially structured generation;
>
> (2) **Hierarchical consistency**: The structure-to-detail nature of next-scale generation in VAR naturally aligns with Haar’s hierarchical LL → LH/HL/HH decomposition, in contrast to the less localized zig-zag ordering used in DCT.
>
> (3) **Computational simplicity**: Haar wavelets are computationally lightweight and easy to integrate.
>
> Overall, Haar-DWT offers aligned coordinates, hierarchical structure, and low overhead, making it a practical fit for our frequency-aware training design. We will include this discussion in the paper.
>
>  ---
>
> **Paper Formatting Concerns**: We thank the reviewer for pointing out the typos and the formatting issue with Figure 3. We will correct these issues in the paper.
>
> ---
>
> **Reference**
>
> [1] Zhang B, Gu S, Zhang B, et al. Styleswin: Transformer-based gan for high-resolution image generation[C]//Proceedings of the IEEE/CVF conference on computer vision and pattern recognition. 2022: 11304-11314.
>
> [2] Yang X, Zhou D, Feng J, et al. Diffusion probabilistic model made slim[C]//Proceedings of the IEEE/CVF Conference on computer vision and pattern recognition. 2023: 22552-22562.
>
> [3] Huang T, You S, Wang F, et al. Knowledge distillation from a stronger teacher[J]. Advances in Neural Information Processing Systems, 2022, 35: 33716-33727.
>
> [4] Jin Y, Wang J, Lin D. Multi-level logit distillation[C]//Proceedings of the IEEE/CVF Conference on Computer Vision and Pattern Recognition. 2023: 24276-24285.
>
> [5] Han J, Liu J, Jiang Y, et al. Infinity: Scaling bitwise autoregressive modeling for high-resolution image synthesis[C]//Proceedings of the Computer Vision and Pattern Recognition Conference. 2025: 15733-15744.
>
> [6] Torralba A, Oliva A. Statistics of natural image categories[J]. Network: computation in neural systems, 2003, 14(3): 391.

---

> > ### Comment · Reviewer_NPSH · 2025-08-08
> > **Keeping the score**
> >
> > Some of my concerns were addressed in the rebuttal by the authors, I am keeping my score.

---

> ### Comment · Area_Chair_ACNy · 2025-08-06
>
> Dear Reviewer NPSH,
>
> This is a reminder that the author-reviewer discussion period is about to close.
>
> Could you please provide your comments on the author rebuttal? We would appreciate your contribution to the discussion.

---

### Official Review · Reviewer_NKKY · 2025-07-05

**Clarity:** 3
**Significance:** 3
**Originality:** 3
**Rating:** 5
**Confidence:** 4

**Summary:**

The paper introduces FreqExit, a methodology that allows dynamic inference Visual Autoregressive (VAR) models. VAR models are characterized by generating images in a coarse-to-fine, next-scale manner. Therefore, existing early-exit strategies fail in this setting due to unstable intermediate representations and the progressive existense of high-frequency details in later steps. The authors achieve dynamic inference with early exit by combining three key strategies:  1) Curriculum-Based Early-Exit Supervision, 2) Progressive High-Frequency Consistency Loss, and 3) A Lightweight Frequency-Gated Self-Reconstruction Loss (FGSR). The first strategy improves the performance of shallow layers. It ensures that all layers are trained to be predicted under supervision with dropout and KD loss relying on layer-wise supervision, depth-aware layer dropout, a rotational curriculum schedule, and a combination of cross entropy and KD for early exit losses. The second techniques aligns intermediate outputs with ground truth in the frequency domain over time. Early generation steps emphasis low-frequency while later steps add high frequency details. This encourages the model to learn frequency-consistent features across steps without disrupting early training dynamics.The FGSR loss tries to enhance frequency learning  via a lightweight auxiliary path using wavelet gates. It acts as an auxiliary loss to guide spectral learning. Empirical evaluation on ImageNet shows that by combining these techniques, FreqExit is able to stabilizes training, support early exists and maintain high fidelity in visual generation especially for models that have coarse-to-fine autoregressive generation like VAR. Overall, FreqExit is able to save computational cost without any architectural modifications and compromising model quality ( 2x inference speedup with negligible quality degradation).

**Questions:**

1- What is the inference-time criterion for triggering early exits? Is it confidence-based or entropy-based? The paper proposes early exist during generation but no clear explanation when to stop at an intermediate layer during inference. In existing early-exit methods, common heuristics include: entropy threshold, confidence threshed or margin criterion (e,g., exit when the gap between top-1 and top-2 logits exceeds a threshold). It would strengthen the paper if the authors clearly explain the exist policy at inference and evaluate it sensitivity and reliability.
2- FGSR module appears to be a key component for stabilizing training. How sensitive is the final performance to the hyperparameter λ, which balances the main task loss with the reconstruction guidance? do you have a sensitivity analysis?
3- The paper shows very good results on the family of VAR models. How generalizable is the FreqExit approach? Could it be applied to other hierarchical or next-scale generative models, or in other modalities such as audio, video or text,  that exhibit similar coarse-to-fine generation dynamics?
4- The early-exit strategies outlined in Table 2 are manually predefined schedules. Have you considered learning an adaptive policy that determines the exit layer for each generation step based on properties of the input image or the intermediate representation? This could potentially lead to even better efficiency-fidelity trade-offs.
5- The paper doesn't provide runtime savings benchmarks. For example FLOPS, latency for the different exit schedules. It would be good to show layer-wise FLOP savings, wall-clock latency improvements, and quality degradation as a function of easy exist. The results only show the speedup. I recommend including a plot showing performance vs. average depth or latency and reporting FLOPS or latency for several early exit schedules.
6- Could the shared LM head limit the informational/representational diversity? There is no analysis or ablation of shared vs. per-later heads. It would be good to compare shared vs. separate heads for shallow and deep layers. Then analyze if deep leaders still learn richer presentations or get regularized.

**Ethical Concerns:**

["NO or VERY MINOR ethics concerns only"]

**Limitations:**

Yes, the authors have adequately addressed the limitations in the NeurIPS checklist included with the paper.

**Quality:**

4

**Strengths And Weaknesses:**

Strengths:
- This work is the first to implement the early-exist strategy in dynamic inference for VAR models.
- The limitations of existing early-exit approaches are well explained and motivated. The empirical analysis of VAR's generation dynamics using DWT (Sec 3.1)  clearly illustrates the problem of frequency progression and layer instability
- The paper is well written and provides good illustrations in Figure 1 to explain the overall methodology. The 3 proposed techniques are well-designed, with each component directly targeting an identified weakness of the baseline model.
- The methodology used is novel and provides a principles approach to supervise models in the frequency space where high-frequency details often correlated with semantic correctness. The combination of early-exist loss, curriculum scheduling and temperature-aware distillation is well-motivated and has sound intuition.
- The experiments are thorough, with comparisons to relevant baselines (CALM, LayerSkip, CoDe)
- The ablation study in Table 3 validates the contribution of the proposed HF loss and FGSR modules.

Weaknesses:
- The scope of evaluation is limited to ImageNet 256x256. While the presented results are strong, the benefits of dynamic inference and especially inference acceleration are more profound at high resolution where the the computational needs are significantly higher.
- The comparison with CALM required modifying the threshold to enable it to work. This kid of indicates the incompatibility of this baselines. It would have been better to frame this as a limitation focusing on why existing methods fail.
- The mathematical explanation of the FGSR module was hard to follow especially the derivation of the closed form update in Equation 9. It would be good if the authors can simplify the explanation in the main body.
- Minor detail: HH, LH, etc symbols are not defined. They might be obvious for some readers as high-frequency bands. It would be better to define them for clarity.
- The inference-time criterion's for triggering early exists is not well explained in the paper
- The paper doesn't provide runtime benchmarks or latency vs. accuracy tradeoff figures.
- The shared LM head across layers could introduce informational or representational bottlenecks. This is not explained or discussed in the paper.
- No deep evaluation of the impact of early exist on the quality of the generation of multi-token contexts.
- No discussion on how this can be generalized to no visual tasks such as audio, video or NLP.

---

> ### Author Rebuttal · Authors · 2025-07-31
>
> We sincerely thank the reviewer for the constructive and insightful feedback. We greatly appreciate your recognition of the strengths of our work and your valuable suggestions. We address each point in detail below.
>
> ---
>
> **W1: Lack of High-Resolution Experiments**
>
> **W1-Ans:** We thank the reviewer for the insightful comment. We regret that we are unable to provide high-resolution results during the rebuttal period. Training at 512×512 resolution is prohibitively expensive. Running 80 epochs with VAR-d36 would take about **24 days on 8×A100 GPUs** (approximately 23% of the original VAR-d36 training time on A100 GPUs). We plan to include high-resolution experiments in the revision.
>
> Notably, **FreqExit is resolution-agnostic**: it adjusts layer depth per generation step without modifying patching or tokenization. Although higher resolutions produce larger token maps, the layer-wise structure within each step remains unchanged, allowing seamless application of our method. Moreover, our baseline **CoDe** is also only evaluated at **256×256** resolution.
>
> Finally, recent work such as **Infinity** [1] adopts the same next-scale generation paradigm and per-step Transformer structure at resolutions beyond 1024, further supporting the generalizability of FreqExit to high-resolution settings.
>
> ---
>
> **W2: Limitations of CALM under Next-Scale Generation**
>
> **W2-Ans:**  We evaluated CALM-style exit criteria (*Softmax Max*, *Diff*, and *Entropy*) on VAR-d16 and VAR-d20 (see Table below), and found that all metrics remain far below typical thresholds, indicating the failure of confidence-based signals to determine viable exits.
>
> **Table 1: Layer-wise CALM Confidence Metrics Comparison**
>
> *Softmax Max = probability of the top-1 token; Softmax Diff = difference between top-1 and top-2 probabilities*
>
> | **Layer** | **Softmax Max(d-16)** | **Softmax Diff** | **Entropy** | **Softmax Max(d-20)** | **Diff** | **Entropy** | **Exit Threshold** |
> | --- | --- | --- | --- | --- | --- | --- | --- |
> | 10 | 0.070 | 0.034 | 6.02 | 0.045 | 0.020 | 6.50 | No |
> | 12 | 0.027 | 0.010 | 6.91 | 0.018 | 0.006 | 6.23 | No |
> | 14 | 0.030 | 0.011 | 6.75 | 0.010 | 0.003 | 6.55 | No |
> | 16 | -- | -- | -- | 0.005 | 0.001 | 6.86 | No |
> | 18 | -- | -- | -- | 0.034 | 0.013 | 5.61 | No |
>
> This is due to CALM’s core assumption that **representations stabilize across layers**, which does not hold in VAR. As next-scale prediction introduces high-frequency details in later steps, each layer continues to substantially update token representations. This leads to large inter-layer changes, also reflected in the low cosine similarity shown in Fig. 2(c).
>
> These results will be included in the revision.
>
> ---
>
> **W3:  FGSR Closed-Form Derivation**
>
> **W3-Ans:** We will revise the main text to improve the explanation of the FGSR module and simplify the derivation of the closed-form update in Eq. (9).
>
> ---
>
> **W4: Lack of Definition**
>
> **W4-Ans:** All frequency band notations (e.g., HH, LH) will be defined upon first use to ensure clarity. Thanks for pointing out the issue!
>
> ---
>
> **W5: Early Exit Strategy Clarification**
>
> **W5-Ans:** We report typical exit paths to enable comparison across typical Latency/FLOPs–FID trade-offs, serving as consistent baselines for evaluating inference cost and quality.
>
> ---
>
> **W6: Lack of benchmark Figures**
>
> **W6-Ans:** We will include visual plots showing performance vs. average depth and latency in the revised version. These cannot be included in the rebuttal per the NeurIPS rebuttal guidelines.
>
> ---
>
> **W7: Discussion of Shared LM Head**
>
> **W7-Ans:** We thank the reviewer for raising this point and are glad to clarify. Sharing the LM head promotes consistent representations and serves as a form of regularization. It also avoids significant parameter overhead. Adding separate heads from layer 10 onward increases parameters by 23.8%–29.9% (see Table 2 below) without improving performance.
>
> **Table 2: Overhead from adding individual LM heads to intermediate layers starting at layer 10 (excluding the final layer).**
> | **Model** | **Added Params** | **Param Overhead (%)** |
> | --- | --- | --- |
> | VAR-d16 | 73.7M | 23.8% |
> | VAR-d20 | 151.3M | 25.2% |
> | VAR-d24 | 287.2M | 27.8% |
> | VAR-d30 | 601.2M | 29.9% |
>
> ---
>
> **W8: Impact of Early Exit on Multi-Token Generation**
>
> **W8-Ans:** We thank the reviewer for raising this important concern. FreqExit adjusts inference depth but **does not alter the underlying autoregressive generation scheme**. The reported FID reflects the quality of full images, thereby capturing coherence across token sequences. Moreover, Fig.4 in the appendix (Zero-Shot Inpainting) shows that **local consistency is well preserved**.
>
> ---
>
> **W9: Generalization of FreqExit**
>
> **W9-Ans:** FreqExit targets next-scale autoregressive visual generation (e.g., image synthesis, inpainting) and already offers practical, resource-aware inference within vision. Because it operates purely at the architectural level, the same depth-adaptive strategy can transfer to hierarchical coarse-to-fine generators in NLP, audio, and video without changing the core training scheme.
>
> ---
>
> **Q1: Inference-Time Exit Criterion**
>
> **Q1-Ans:** In our current paper, **early exits follow fixed depth paths chosen offline**. We benchmark several representative paths and report their **speed (latency / FLOPs) and quality (FID)** so readers can make direct trade-off comparisons. No confidence or entropy threshold is applied at inference time. These fixed paths serve as practical heuristics and reproducible baselines for typical deployment budgets. In future work, we will explore adaptive per-sample exit rules, such as confidence signals or lightweight policy networks, to further improve flexibility.
>
> ---
>
> **Q2: Sensitivity Analysis of λ**
>
> **Q2-Ans:** FGSR uses two mechanisms for stable training: (i) a learnable sub-band gate γ_b and (ii) an auxiliary reconstruction loss. The hyper-parameter λ scales only the orthogonal alignment term L_align, regularizing sub-band projections **without perturbing the main gradients**. We have added a sensitivity analysis on VAR-d16 that varies λ and confirms training remains stable across a broad range of values.
>
> **Table 3** reports FID scores at two checkpoints (Epoch 8 and Epoch 16) for three λ values and three early-exit layers.
> | λ | L10 (Epoch 8) | L12 (Epoch 8) | L14 (Epoch 8) | L10 (Epoch 16) | L12 (Epoch 16) | L14 (Epoch 16) |
> | --- | --- | --- | --- | --- | --- | --- |
> | 0.05 | 26.8 | 20.2 | 7.31 | 18.3 | 12.6 | 6.26 |
> | 0.10 | 26.8 | 20.2 | 7.45 | 18.3 | 12.7 | 6.24 |
> | 0.20 | 27.1 | 20.0 | 7.37 | 18.1 | 12.6 | 6.16 |
>
> The differences across λ in [0.05, 0.20] are marginal, indicating that FGSR is **insensitive** to reasonable choices of λ. We therefore set λ = 0.1.
>
> ---
>
> **Q3: Applicability of FreqExit Beyond VAR**
>
> **Q3-Ans:** FreqExit is an architectural mechanism: every layer learns to emit a full sample, and inference depth is selected on demand. This layer-centric design is **token-agnostic and thus transferable**. We have already confirmed its effectiveness on visual generation with VAR models. We believe the same depth-adaptive scheme could be applied to other hierarchical or next-scale generators, including those in NLP, audio, and video. We will explore these modalities in future work.
>
> ---
>
> **Q4: Adaptive Early-Exit Strategy**
>
> **Q4-Ans:** We agree this is a valuable suggestion. FreqExit presently uses a few fixed exit paths to cover typical latency versus FID trade offs. Learning a per sample exit rule by using confidence cues is a promising next step. We plan to investigate it in future work.
>
> ---
>
> **Q5: Runtime Savings Benchmarks**
>
> **Q5-Ans:** We clarify that Table 1 reports latency, throughput, and GFLOPs together with FID for each exit schedule, providing a comparison of speed-quality trade offs. To make these trends clearer, We will include a performance vs. latency plot in the revision.
>
> ---
>
> **Q6: Impact of  Shared and Sperate LM Heads**
>
> **Q6-Ans:**  A shared LM head keeps representations consistent, serves as implicit regularization, and avoids the extra parameters incurred by per-layer heads. As shown in Table 2, adding separate heads from layer 10 onward would raise parameters by roughly 24%–30%. Consistent with prior work [2-3], we therefore keep a single shared head.
>
> ---
>
> **Reference**
>
> [1] Han J, Liu J, Jiang Y, et al. Infinity: Scaling bitwise autoregressive modeling for high-resolution image synthesis[C]//Proceedings of the Computer Vision and Pattern Recognition Conference. 2025: 15733-15744.
>
> [2] Elhoushi M, Shrivastava A, Liskovich D, et al. LayerSkip: Enabling early exit inference and self-speculative decoding[J]. arXiv preprint arXiv:2404.16710, 2024.
>
> [3] Liu F, Tang Y, Liu Z, et al. Kangaroo: Lossless self-speculative decoding for accelerating llms via double early exiting[J]. Advances in Neural Information Processing Systems, 2024, 37: 11946-11965.

---

### Note · Authors · 2025-08-11

We thank the AC and all reviewers for their constructive feedback and recognition during the discussion phase. Their insightful reviews have greatly contributed to improving the clarity and overall quality of our work.

Reviewer **NKKY** (**Initial Rating:5, Confidence:4**) evaluated our work as *“the first to implement the early-exit strategy in dynamic inference for VAR models”* and noted that *“the methodology is novel.”* The reviewer recognized the contribution of curriculum-based early-exit supervision, high-frequency consistency, and FGSR in improving shallow-layer quality. We further provided theoretical clarifications and quantitative evidence during the discussion to reinforce its practicality and generality.

Reviewer **NPSH** (**Initial Rating:4, Confidence:3**) appreciated our suite of techniques for accelerating VAR modeling and recognized the role of curriculum-based early-exit supervision, high-frequency consistency, and FGSR. The reviewer valued our discussion-phase additions on frequency progression and wavelet-domain design, confirming the method’s soundness and applicability.

Reviewer  **sjr1** (**Initial Rating:4, Confidence:4**) described our work as *“a solid paper”* and recognized the effectiveness of our early-exit framework for next-scale VAR. The reviewer valued the additional ablations on depth-aware weighting and progressive temperature scheduling, acknowledging their distinct contributions to improving early-exit performance.

Reviewer **paNB** (**Initial Rating:3, Confidence:3**)  acknowledged our clear motivation for training-based early-exit supervision, appreciated the strengthened quantitative evidence, and recognized the method’s adaptability to both next-scale and next-token VAR settings through our extended evaluations. The reviewer expressed satisfaction with the clarifications and stated he/she **will raise the score**.

**FreqExit** delivers the first effective early-exit–based dynamic inference for next-scale VAR, offering strong acceleration–quality trade-offs without architectural changes. Its validated design and robustness make it a valuable contribution with broad potential impact. We sincerely thank the reviewers for recognizing these strengths and their positive feedback on the clarity, novelty, and efficacy of our method. Their comments motivate us to refine and strengthen it based on the points raised, and we are committed to addressing these concerns with a detailed response during the rebuttal.

---

### Decision · Program_Chairs · 2025-09-17

**Decision:**

Accept (poster)

**Comment:**

This paper introduces FreqExit, a novel and well-motivated framework for enabling efficient early-exit inference in Visual Autoregressive (VAR) models. By identifying that VARs' coarse-to-fine generation process violates the assumptions of standard dynamic inference methods—specifically through the late emergence of high-frequency details—the authors propose a technically sound, frequency-aware supervision strategy. The work received a consensus for acceptance (final scores: 5, 4, 4, 4) following a highly constructive discussion phase.

The recommendation to accept is based on the paper's novelty in addressing an unsolved problem and its strong technical execution. Reviewers unanimously recognized the insightful analysis in the work and found the proposed method to be effective, demonstrating significant inference speedups. Initial concerns regarding motivation, the contribution of individual components, and the paper's positioning were thoroughly resolved through rebuttal and discussion. The authors provided detailed clarifications, new ablation studies during the review period, and committed to refining the paper's claims, which was instrumental in convincing initially borderline reviewers.

While the evaluation is limited to a single dataset and resolution, this is an acceptable limitation given the high computational cost of training these models. The engagement during the review process has substantially improved the paper and solidified its contribution.